# Local mitochondrial physiology defined by mtDNA quality guides purifying selection

Felix Thoma[1,2], Johannes Hagen[1,2☿], Romina Rathberger[1,2☿],
Francesco Padovani[3], David Hörl[1,4], Kurt M. Schmoller[3], Christof Osman[1]*

**1** Faculty of Biology, Ludwig-Maximilians-Universität München, Planegg-Martinsried, Germany,
**2** Graduate School Life Science Munich, Ludwig-Maximilians-Universität München, Planegg-Martinsried, Germany, **3** Institute of Functional Epigenetics, Molecular Targets and Therapeutics Center, Helmholtz Zentrum München, Neuherberg, Germany, **4** Center for Molecular Biosystems (BioSysM), Human Biology and BioImaging, Ludwig-Maximilians-Universität München, Munich, Germany

☿ These authors contributed equally to this work.
* osman@bio.lmu.de

## Abstract

The mitochondrial genome (mtDNA) encodes essential subunits of the electron transport chain and ATP synthase. Mutations in these genes impair oxidative phosphorylation, compromise mitochondrial ATP production and cellular energy supply, and can cause mitochondrial diseases. These consequences highlight the importance of mtDNA quality control (mtDNA-QC), the process by which cells selectively maintain intact mtDNA to preserve respiratory function. Here, we developed a high-throughput flow cytometry assay for *Saccharomyces cerevisiae* to track mtDNA segregation in cell populations derived from heteroplasmic zygotes, in which wild-type (WT) mtDNA is fluorescently labeled and mutant mtDNA remains unlabeled. Using this approach, we observe purifying selection against mtDNA lacking subunits of complex III (*COB*), complex IV (*COX2*) or the ATP synthase (*ATP6*), under fermentative conditions that do not require respiratory activity. By integrating cytometric data with growth assays, qPCR-based mtDNA copy-number measurements, and simulations, we find that the decline of mtDNA$^{\Delta atp6}$ in populations derived from heteroplasmic zygotes is largely explained by the combination of its reduced mtDNA copy number—biasing zygotes toward higher contributions of intact mtDNA—and the proliferative disadvantage of cells carrying this variant. In contrast, the loss of mtDNA$^{\Delta cob}$ and mtDNA$^{\Delta cox2}$ cannot be explained by growth defects and copy-number asymmetries alone, indicating an additional intracellular selection against these mutant genomes when intact mtDNA is present. In heteroplasmic cells containing both intact and mutant mtDNA, fluorescent reporters revealed local reductions in ATP levels and membrane potential ($\Delta\Psi$) near mutant genomes, indicating spatial heterogeneity in mitochondrial physiology that reflects local mtDNA quality. Disruption of the respiratory chain by deletion of nuclear-encoded subunits (*RIP1*, *COX4*) abolished these physiological gradients and impaired mtDNA-QC, suggesting that local bioenergetic differences are required

**Data availability statement:** All cropped images used for data quantification in the figures are accessible here: https://osf.io/h63ca/overview (main figures) https://osf.io/xmh8b/overview (supplementary) The full raw microscopy data exceed 1 TB and were therefore not uploaded to an online repository; however, they can be made available upon request.

**Funding:** This work was supported by grants of the Deutsche Forschungsgemeinschaft (https://www.dfg.de/) awarded to CO (OS 410/3-1) and DH (HO 7333/1). The work was also supported by a grant of the Human Frontier Science Program (https://www.hfsp.org/) awarded to CO (GP021/2023). The funders had no role in study design, data collection and analysis, decision to publish, or preparation of the manuscript.

**Competing interests:** The authors have declared that no competing interests exist.

for selective recognition. Together, our findings support a model in which yeast cells assess local respiratory function as a proxy for mtDNA integrity, enabling intracellular selection for functional mitochondrial genomes.

## Author summary

Mitochondria are essential organelles in our cells that convert nutrients into usable cellular energy. They contain their own DNA, and mutations in this DNA can compromise mitochondrial function and contribute to disease. In this work, we asked how cells recognize and limit the transmission of defective mitochondrial DNA. Using baker's yeast as a model system, we developed a rapid approach that allows us to track how healthy and mutant mitochondrial DNA variants are inherited when cells divide. We found that cells preferentially retain mitochondrial DNA that supports normal mitochondrial function, even under conditions where respiration is not required. By combining cell sorting, growth measurements, microscopy, and simulations, we show that this selective process is influenced not only by the initial abundance of each mitochondrial genome, but also by localized physiological differences within the mitochondrial network. Regions containing mutant mitochondrial DNA exhibit reduced membrane potential and lower ATP availability, making them distinguishable from regions harboring intact genomes. When we impaired the respiratory chain, these physiological differences disappeared and selective quality control broke down. Our findings suggest that cells rely on local mitochondrial signals to preserve functional mitochondrial DNA, offering insight into how healthy mitochondria are maintained across generations.

## Introduction

Mitochondria contain their own genome, mitochondrial DNA (mtDNA), which is present in multiple copies in each cell. mtDNA encodes a limited but critical subset of mitochondrial proteins, which are integral subunits of the respiratory chain. Mutations in these genes often result in respiratory dysfunction, contributing to the pathogenesis of a wide range of diseases [1–3]. Purifying selection, which has been shown to act in the germline of mammalian cells and *Drosophila melanogaster*, acts to preserve mitochondrial DNA (mtDNA) integrity, by preventing maintenance of mutant mtDNA copies [4–9].

Our previous work demonstrated that even the single-celled yeast *Saccharomyces cerevisiae* can distinguish between intact and mutant mtDNA, establishing it as a powerful model for studying mitochondrial DNA quality control (mtDNA-QC) [10]. Remarkably, this quality control operates within a fused, continuous mitochondrial network and does not require the fission protein Dnm1. We further showed that selection against mutant mtDNA depends on intact mitochondrial cristae, which limit the diffusion of mtDNA-encoded proteins and thus preserve a spatial link between an mtDNA molecule's genotype and its encoded proteins. These findings support the "sphere of influence" model [10,11], which proposes that cristae-mediated

compartmentalization of the mitochondrial inner membrane enables the emergence of distinct local physiological states that reflect the underlying mtDNA quality. Such spatial heterogeneity may in turn enable selective recognition and intracellular selection against defective genomes. However, whether mutant mtDNA can indeed generate localized physiological differences within a continuous mitochondrial network that also harbors intact genomes has remained a critical open question.

To date, our investigations have primarily focused on a competitive context in which WT mtDNA coexists with mutant mtDNA lacking the *COB* gene, which encodes a core subunit of complex III [10,12]. However, it remains unclear whether mutations affecting subunits of complex IV or the ATP synthase are similarly subject to purifying selection. This question is relevant because defects in different components of the respiratory chain have distinct bioenergetic consequences. Loss of complexes III or IV compromises the electron transport chain's ability to generate a proton gradient, requiring reverse operation of the ATP synthase to maintain the membrane potential ($\Delta\Psi$), a process that consumes ATP and likely leads to ATP depletion [13–15]. In contrast, loss of the mtDNA-encoded Atp6, a $F_O$ subunit of the ATP synthase, does not directly impair proton pumping by complexes III and IV. Because protons can no longer re-enter the matrix via the ATP synthase in the absence of Atp6, dissipation of $\Delta\Psi$ is impaired, potentially resulting in membrane hyperpolarization, while ATP synthesis is abolished [16,17] . However, the absence of ATP synthase has been reported to cause assembly defects in complex IV, further compounding bioenergetic dysfunction [18,19]. Moreover, since it is part of the ATP synthase, mutation or absence of Atp6 also affects ATP-synthase assembly dimer formation and thereby alters cristae morphology [18] with potential consequences for the "sphere of influence" of individual mtDNA copies. Taken together, these observations suggest that the physiological consequences of specific mtDNA mutations may differ, and how these differences influence purifying selection remains an open question.

In this study, we developed a rapid analysis pipeline, termed FAST (**F**low cytometry **A**nalysis for **S**egregation **T**racking), which combines micromanipulation with flow cytometry to monitor the segregation of WT and mutant mtDNA variants over multiple generations. Using FAST, we analysed the competitive dynamics between intact mtDNA and mutant genomes lacking genes encoding subunits of complex III, complex IV, or the ATP synthase in populations derived from heteroplasmic zygotes containing both genome types. We further investigated whether $\Delta\Psi$ and ATP levels differ locally within the mitochondrial network depending on the underlying mtDNA genotype. Finally, we tested whether purifying selection against mutant mtDNA requires a functional respiratory chain by assessing whether global impairment of oxidative phosphorylation disrupts the selective retention of intact mtDNA.

## Results

### Development of a rapid pipeline for mtDNA segregation analysis

In the following, we describe the development of a novel flow cytometry–based pipeline that enables quantitative analysis of mtDNA variant segregation in cell populations derived from single heteroplasmic zygotes carrying two distinct mitochondrial genomes. This approach builds on previous microscopy-based strategies that employed heteroplasmic yeast zygotes containing mtDNA encoding the fluorescent fusion protein Atp6-mNeonGreen (mtDNA$^{ATP6-NG}$) alongside a non-fluorescent mtDNA ("dark" mtDNA) [12].

First, we asked whether flow cytometry allows reliable discrimination between non-fluorescent cells and cells harboring mtDNA$^{ATP6-NG}$. To this end, we analysed three distinct populations: (i) 'dark' cells containing mtDNA not encoding fluorescent proteins, (ii) mtDNA$^{ATP6-NG}$ cells and (iii) a mixture of both (Fig 1A–1C). Flow cytometric analysis revealed that mtDNA$^{ATP6-NG}$ cells could be readily distinguished from dark cells. Although non-fluorescent and mtDNA$^{ATP6-NG}$ cells exhibited similar cell size based on forward scatter (FSC-A), only the latter showed markedly increased fluorescence intensity. Approximately 1.9% of cells from the mtDNA$^{ATP6-NG}$ strain lacked detectable fluorescence (S1A Fig). These likely correspond to petite cells that have partially or completely lost mtDNA including the Atp6-NG locus or elements required for mitochondrial translation. Notably, this proportion is lower than the ~5% petite frequency typically observed

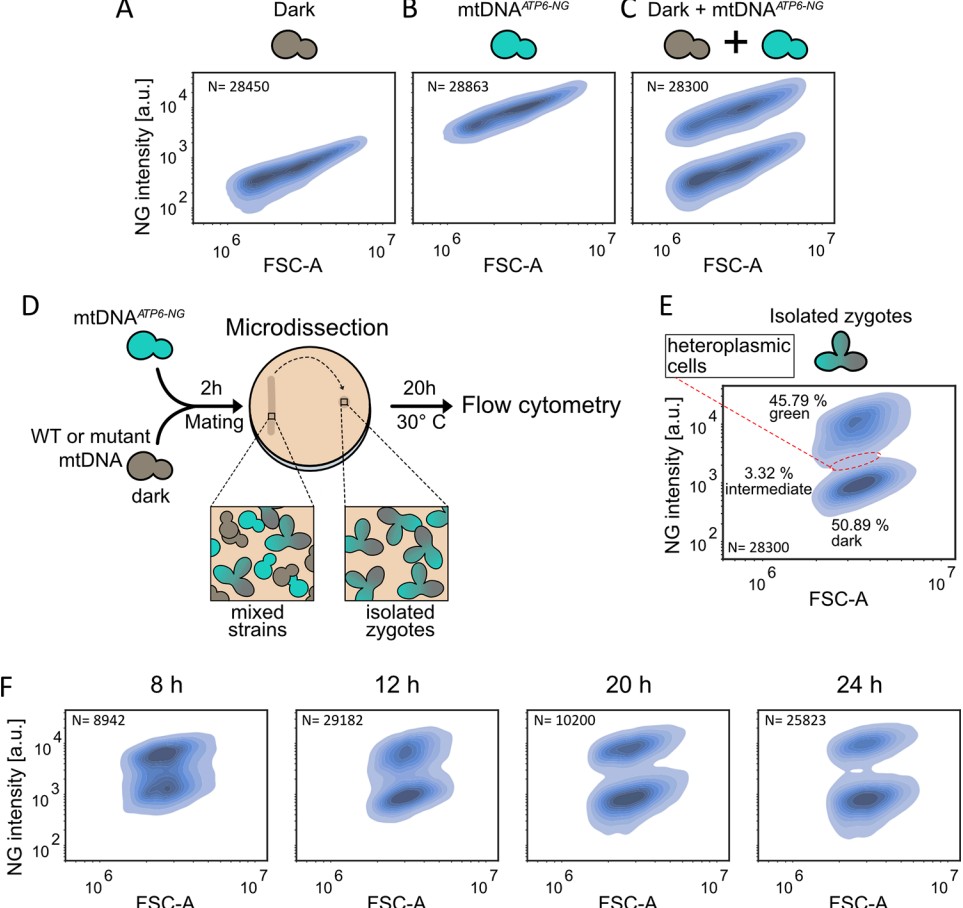

**Fig 1. Flow cytometry based method to assess mtDNA quality control.** Flow-cytometric density plot of yeast cells lacking NG **(A)**, containing mtDNA$^{ATP6-NG}$ **(B)**, and a mixed population of both strains **(C)** with fluorescence intensity on the y-axis and forward scatter (FSC-A) on the x-axis. Note that non-fluorescent cells are not visible in (B) due to their low abundance. **(D)** Schematic of the FAST experimental workflow. **(E)** FAST density plot of mtDNA$^{ATP6-NG}$ × mtDNA$^{ic}$ heteroplasmic microcolonies after 20 h: the area highlighted with a red dashed circle indicates heteroplasmic cells; percentages of fluorescent, heteroplasmic, and non-fluorescent populations are annotated next to each cluster. **(F)** mtDNA$^{ATP6-NG}$ cells were mated with mtDNA$^{ic}$ cells. 310, 150, 50, and 40 zygotes were isolated to discrete coordinates on an agar plate and microcolonies were analysed 8, 12, 20, and 24 h post-mating, respectively.

in our background. A plausible explanation is that some petite cells retain the *ATP6-NG* reporter locus while harboring mutations or deletions elsewhere in the mtDNA that render them respiration-deficient. Such cells would remain fluorescent despite being petite, thereby reducing the apparent fraction of non-fluorescent cells. The difference between mtDNA$^{ATP6-NG}$ and dark cells was particularly evident in the mixed population, where the two cell types formed clearly separated clusters (Fig 1C).

In the next step, we tested whether flow cytometry enables rapid and quantitative analysis of mtDNA segregation from heteroplasmic founder cells. On glucose containing medium we crossed the mtDNA$^{ATP6-NG}$ (intron-containing) strain with a WT strain harboring intact intron-containing mtDNA (mtDNA$^{ic}$), which lacks a gene encoding a fluorescent protein, to generate heteroplasmic zygotes. Note that, depending on the experimental context, both intron-containing and intron-less mtDNA (mtDNA$^{il}$) variants are used throughout this study. After zygote formation on agar plates, we isolated 30 zygotes using a microdissection microscope and transferred them to a single separate area on the plate. The plates were then cultured for 20 hours to allow microcolony formation, after which the colony was collected and subjected to flow

cytometry analysis (Fig 1D). Flow cytometry revealed clear mtDNA segregation, with 45.8% of cells displaying fluorescence and 50.9% showing no fluorescent signal (Fig 1E). Notably, the size distribution in Fig 1E is narrower than in the liquid-grown controls (Fig 1A–1C), likely because the microcolonies came from cells grown on solid medium, while the controls were from log-phase liquid cultures, which show more size variation. While the segregation of cells into two clusters was evident, a subset of cells displayed intermediate fluorescence (3.32%, highlighted in Fig 1E). Those intermediate cells were defined based on a narrow fluorescence intensity window (±10% of the distance between peaks) centered around the local minimum between the two major peaks in the kernel density estimate (KDE) of the NG signal distribution (S1D Fig). We hypothesized that these might represent residual heteroplasmic cells. To explore transition from heteroplasmic to homoplasmic cells, we monitored the cell population over a time course at 8, 12, 20, and 24 hours after mating. Indeed, we observed a gradual shift toward two virtually completely separated discrete clusters: one fully fluorescent and one non-fluorescent (Fig 1F).

Atp6-NG may persist even after loss of the mtDNA$^{ATP6-NG}$ by which it is encoded, which could lead to an overestimation of the fraction of cells containing mtDNA$^{ATP6-NG}$. To assess the extent of this effect, we quantified the decrease in Atp6-NG fluorescence per cell over time after inhibiting mitochondrial translation with chloramphenicol. From this analysis, we estimate an apparent fluorescence half-life of 2.7 h in cells, which likely reflects a combination of dilution by cell division and protein turnover. Comparison of Atp6-NG signal intensities after 2, 4, 6, and 8 hours of chloramphenicol treatment with untreated controls revealed a progressive decrease, with fluorescence clearly reduced and readily distinguishable from untreated cells by 4 hours (S1C Fig). These data indicate that only cells that have lost mtDNA$^{ATP6-NG}$ within approximately the last two hours and hence the previous cell cycle would still appear Atp6-NG–positive in our assays. Furthermore, as shown previously, populations derived from heteroplasmic founder cells segregate almost completely within 20 hours [12]. Hence, although a minor residual signal cannot be fully excluded, the stability of Atp6-NG is unlikely to have a substantial impact on our FAST quantifications.

Our previous research had demonstrated that the first cell division of a zygote plays a major role in the segregation of mtDNA variants [10]. This led us to test if segregation might be faster when analysing populations derived from the first daughter cell of a heteroplasmic zygote, rather than populations derived from the zygote itself. To test this, we isolated the first daughter cells from 30 heteroplasmic zygotes, transferred them to a separate location on the agar plate, and examined mtDNA segregation after 20 hours (S1E Fig). This approach indeed revealed a more complete segregation of mtDNA variants compared to the analysis performed directly on the zygotes: the two clusters were more distinct, and the proportion of heteroplasmic cells fell from 3.32% to 1.8% (S1B and S1D Fig). Hence, we applied this procedure for all following mtDNA segregation experiments.

In summary, this workflow enables fast data acquisition and high-throughput, single-cell resolution analysis. We refer to this streamlined approach as FAST- **F**low cytometry **A**nalysis for **S**egregation **T**racking.

## Competitive advantage of intact over mutant mtDNA

We next sought to determine whether mtDNA harboring deletions of different genes encoding subunits of different respiratory chain complexes would exhibit altered segregation patterns. In our previous studies, we demonstrated a pronounced preference for WT mtDNA inheritance when mating WT strains with strains harboring a deletion of the *COB* gene in their mtDNA (mtDNA$^{\Delta cob}$) using microscopic or genetic readouts [10,12]. Of note, the mtDNA$^{\Delta cob}$ strain was generated from intron-less mtDNA. In intron-containing mtDNA, a *COB* intron encodes a maturase required for *COX1* mRNA splicing. Deleting *COB* in that context would therefore also block *COX1* expression [20]. To control for effects of the intronless nature of mtDNA$^{\Delta cob}$, we included mating experiments as controls in our FAST analysis in which strains containing mtDNA$^{ATP6-NG}$ were crossed with strains containing either intact mtDNA$^{ic}$ or mtDNA$^{il}$ [21](Fig 2A and S1B Fig). Note that all FAST analyses were performed on cells mated and grown on glucose containing medium. We confirmed the purifying selection against Δ*cob* mtDNA : mating WT mtDNA$^{ATP6-NG}$ with mtDNA$^{\Delta cob}$ resulted in a significant skew towards cells

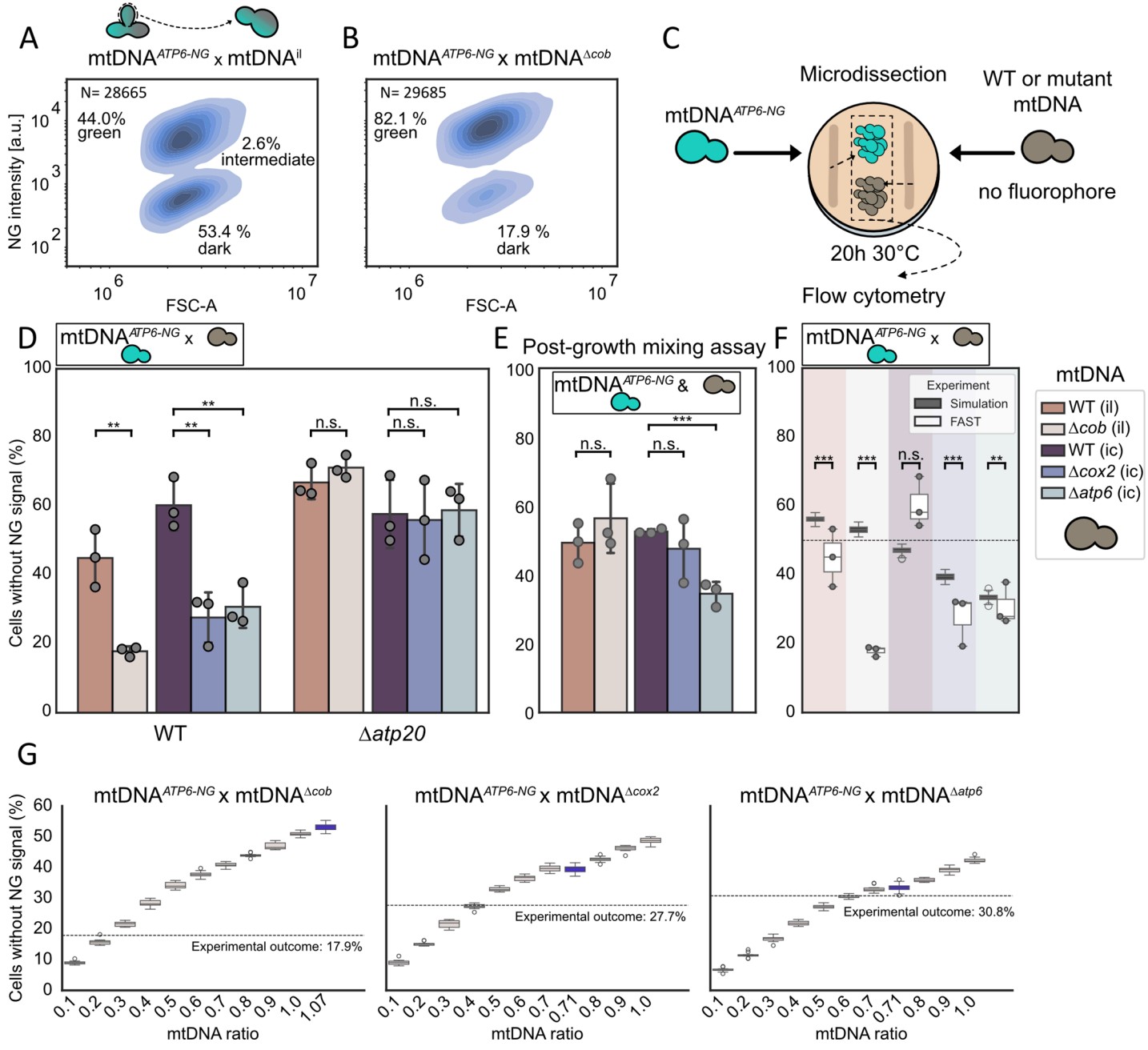

**Fig 2. Competitive advantage of WT over mutant mtDNA.** FAST results for matings of mtDNA$^{ATP6-NG}$ with **(A)** mtDNA$^{il}$ or **(B)** mtDNA$^{\Delta cob}$. 30 first daughter cells from formed zygotes were isolated by micromanipulation, cultured for 20 h, and analysed by flow cytometry. **(C)** Schematic of the post-growth mixing assay workflow: 30 cells of mtDNA$^{ATP6-NG}$ or non-fluorescent strains were microdissected to a defined, separate position, cultured for 20 h, then pooled for flow-cytometric analysis. **(D)** Bar graph depicting the fraction of non-fluorescent cells for each FAST mtDNA$^{ATP6-NG}$ mating combination in WT and $\Delta atp20$ strains. Individual replicates are indicated by dots. **(E)** Results of the post-growth mixing assay showing the proportion of non-fluorescent cells. **(F)** Comparison of FAST and simulation results. In the simulation, mtDNA fate was modeled over time without assuming any selective intracellular advantage for either mtDNA variant. **(G)** Simulation outcomes for mtDNA$^{\Delta cob}$, mtDNA$^{\Delta cox2}$, and mtDNA$^{\Delta atp6}$ matings across a range of initial mtDNA ratios (0.1–1), each simulated 10 times. The experimentally determined mtDNA ratio for each strain was simulated separately (100 iterations as for (F)) and is highlighted in blue. Statistical significance in panels (D) and (E) was determined by unpaired Student's t-test on the means of replicates. Statistical significance in panel (F) was determined by Monte Carlo comparison of each FAST replicate to the simulation distribution, with replicate one-sided p-values combined using Fisher's method to test for a significant reduction of cells containing 'dark' mtDNA in simulations compared to the FAST experiments. (*$P < 0.05$; **$P < 0.005$; ***$P < 0.001$).

containing intact mtDNA, with approximately 82.1% WT mtDNA$^{ATP6\text{-}NG}$-containing cells and only around 17.9% dark cells (Fig 2B). Because FAST relies on competition between intact and mutant mtDNA contributed by the two parental cells, it was important to test whether mtDNA is stably maintained in the mtDNA$^{\Delta cob}$ parent. If mtDNA$^{\Delta cob}$ cells were to become $\rho^0$ or $\rho^-$ at high frequencies, mating them with mtDNA$^{ATP6\text{-}NG}$ cells would alter the competitive dynamics and could bias the FAST outcome. To rule out this possibility, we screened for $\rho^0$ cells by DAPI staining, which did not reveal any cells completely lacking mtDNA (S2H Fig). To test for partial mtDNA loss ($\rho^-$), we plated single mtDNA$^{\Delta cob}$ cells and crossed the resulting colonies to tester strains carrying defined mitochondrial deletions. All 50 mtDNA$^{\Delta cob}$ colonies fully complemented the tester strains, indicating retention of the corresponding mtDNA regions (S2G Fig). Thus, we found no evidence for $\rho^0$ cells or for a substantial fraction of $\rho^-$ cells in the mtDNA$^{\Delta cob}$ strain.

In contrast to the mtDNA$^{ATP6\text{-}NG}$ ×mtDNA$^{\Delta cob}$ cross, mating strains containing mtDNA$^{ATP6\text{-}NG}$ with WT strains harboring intact mtDNA$^{ic}$ or mtDNA$^{il}$ resulted in a more balanced segregation (Fig 2A–2B and S1B Fig). Notably, we observed a slight competitive advantage of mtDNA$^{ic}$ over mtDNA$^{ATP6\text{-}NG}$, resulting in a higher proportion of cells retaining the 'dark' mtDNA$^{ic}$. This suggests a modest selection against mtDNA$^{ATP6\text{-}NG}$ within the arising populations and that Atp6-NG might be functionally compromised, despite the absence of detectable growth defects in strains harboring mtDNA$^{ATP6\text{-}NG}$, which was assessed in previous studies [10].

Next, we asked whether the observed selective disadvantage is specific to the *COB* gene or whether similar selection patterns apply to mutations in other mtDNA-encoded genes. To address this, we extended our analysis to mtDNA deletion strains lacking either *COX2* (mtDNA$^{\Delta cox2}$), which encodes a core subunit of complex IV, or *ATP6*, a core component of the ATP synthase complex (mtDNA$^{\Delta atp6}$) [22–24]. Like the mtDNA$^{\Delta cob}$ strain, mtDNA$^{\Delta cox2}$ and mtDNA$^{\Delta atp6}$ strains were unable to grow on non-fermentable carbon sources (S1F Fig). Moreover, the mtDNA$^{\Delta cox2}$ and mtDNA$^{\Delta atp6}$ strains did not exhibit pronounced loss of mtDNA (S1G and S1H Fig). We then performed matings between strains containing intact mtDNA$^{ATP6\text{-}NG}$ and strains harboring either mtDNA$^{\Delta cox2}$ or mtDNA$^{\Delta atp6}$. Both mtDNA$^{\Delta cox2}$ and mtDNA$^{\Delta atp6}$ contain introns, and thus results from the FAST analyses involving these strains must be compared to those from a mtDNA$^{ATP6\text{-}NG}$ ×mtDNA$^{ic}$ cross. In both cases, the amount of non-fluorescent cells was reduced (27.6% and 30.8%, respectively), pointing to a preferential maintenance of intact mtDNA in the resulting colonies (Fig 2D).

We validated these findings by repeating the analysis using fluorescence microscopy instead of flow cytometry. For this analysis, individual colonies derived from the first daughter cells of heteroplasmic zygotes were imaged separately to assess cell-to-cell variability. This approach highlighted a stochastic element in the segregation process, with some colonies from heteroplasmic zygotes composed almost entirely of non-fluorescent (dark) cells and others containing predominantly fluorescent cells (S2A and S2B Fig). Nonetheless, when results were averaged across multiple colonies, they consistently converged on a reproducible mean, which was comparable to our findings obtained through FAST.

To interpret the outcomes of FAST and to infer modes of selection for or against mtDNA variants, the following two parameters must be taken into account. First, the two parental cells may contribute unequal amounts of each mtDNA variant to the heteroplasmic zygote and bias the predominance of a certain mtDNA variant in the arising population. Second, strains carrying mutant mtDNA variants may exhibit slower growth rates, even on fermentable medium, compared to their corresponding WT strains, potentially contributing to a lower percentage of cells containing mutant mtDNA in populations resulting from heteroplasmic founder zygotes.

First, we tested for putative growth defects associated with the presence of mutant mtDNA variants. Assessment of doubling times for strains containing either intact (mtDNA$^{il}$ or mtDNA$^{ic}$) or mutant (mtDNA$^{\Delta cob}$, mtDNA$^{\Delta cox2}$, or mtDNA$^{\Delta atp6}$) mtDNA in liquid fermentable medium did not reveal significant growth differences (S2C and S2D Fig). However, the FAST analysis is performed on solid medium, where spatially heterogeneous growth conditions within colonies may impose additional metabolic challenges [25–27]. To better reflect these conditions, we sought to assess the growth behavior of the different yeast strains under similar settings, and therefore repeated the FAST analysis without the mating step. Specifically, thirty haploid budding cells were isolated and deposited at a defined position on the agar plate containing glucose as a carbon source, giving rise to a microcolony derived from these founder cells. This procedure

was performed for all strains harboring the various mtDNA variants. After 20 hours of growth, the resulting colonies were harvested and pooled in combinations mimicking those used in the mating-based FAST experiments. The pooled populations were then immediately analysed by flow cytometry. We refer to this assay as 'post-growth mixing assay' (Fig 2C).

For mixtures of WT mtDNA$^{ATP6-NG}$ and either mtDNA$^{\Delta cob}$ or mtDNA$^{\Delta cox2}$ cells, approximately 50% of the population exhibited fluorescence, consistent with similar growth rates between strains (Fig 2E). Thus, the observed reduction of non-fluorescent mutant cells in the FAST experiments involving mtDNA$^{\Delta cob}$ or mtDNA$^{\Delta cox2}$ is unlikely to be caused by growth differences. In contrast, when we performed the post-growth mixing assay with mtDNA$^{ATP6-NG}$ and mtDNA$^{\Delta atp6}$ cells, a lower proportion of mtDNA$^{\Delta atp6}$ cells was observed, indicating a growth disadvantage for this strain under plate-based conditions. This growth defect likely contributes to a decreased percentage of mtDNA$^{\Delta atp6}$ cells in FAST analysis of populations derived from heteroplasmic zygotes.

In our post-growth mixing assay, cells were homoplasmic for either mtDNA$^{ATP6-NG}$ or mtDNA$^{\Delta atp6}$, which is an extreme scenario that would arise if complete segregation of mtDNA would occur during the first cell division. In contrast, mtDNA segregation starts from heteroplasmy and proceeds gradually [12]. Hence, the growth disadvantage due to mtDNA$^{\Delta atp6}$ is likely minor at first and increases as cells approach homoplasmy.

The second factor potentially biasing the segregation experiment is that unequal mtDNA copy numbers in the two parental cells lead to unequal variant ratios in the resulting heteroplasmic zygote. We first quantified mtDNA copy number in strains carrying exclusively intact or mutant mtDNA by quantitative PCR. As a reference, we used the mtDNA$^{ATP6-NG}$ strain that was used in all matings. Unexpectedly, strains harboring mtDNA$^{il}$ displayed elevated mtDNA copy numbers compared to the mtDNA$^{ATP6-NG}$ strain (S2E Fig). Strains containing mtDNA$^{\Delta cob}$ also showed increased mtDNA copy number relative to the mtDNA$^{ATP6-NG}$ strain, although slightly lower than in mtDNA$^{il}$. mtDNA$^{ic}$ strains carried mtDNA amounts comparable to mtDNA$^{ATP6-NG}$, whereas mtDNA$^{\Delta cox2}$ and mtDNA$^{\Delta atp6}$ strains showed a modest reduction.

To better resolve how different mtDNA copy numbers and growth rates influence FAST outcomes, we used a previously established mtDNA-segregation simulation framework [12]. This model simulates mtDNA segregation at the population level and incorporates major features of mitochondrial dynamics, including fusion, fission, and mtDNA partitioning during cell division (S2G Fig). We extended the model by adding a growth-correction factor for heteroplasmic cells containing intact and mutant mtDNA, assuming a linear relationship between the intracellular percentage of each mtDNA variant and its contribution to cellular growth rate.

To parameterize the simulation, we first determined the doubling time of mtDNA$^{ATP6-NG}$ cells on plates containing glucose (1.48 h) by seeding 10, 20, or 30 cells and quantifying cell numbers after 20 hours via flow cytometry (S2F Fig). Combined with the fluorescence ratios obtained in the post-growth mixing assay, these measurements allowed us to calculate the relative growth rate of cells containing mutant mtDNA compared with mtDNA$^{ATP6-NG}$ cells, which served as the growth-correction factors in the model. We also incorporated the measured starting copy-number differences between mtDNA variants. Thus, starting cells were initialized with mtDNA amounts proportional to those found in the corresponding haploid parental strains. For example, in a cross between mtDNA$^{ATP6-NG}$ and mtDNA$^{\Delta cox2}$ parental cells, the starting cell contained a ratio of mtDNA$^{ATP6-NG}$:mtDNA$^{\Delta cox2}$ of 1:0.71.

Using these parameters, the model predicted the proportion of mtDNA$^{ATP6-NG}$ in the population after ~20 h of growth, mirroring the conditions of the FAST assay. Importantly, the simulations do not assume any intracellular disadvantage of mutant mtDNA; they incorporate only growth-rate differences and the measured initial mtDNA abundances. Given the lack of a growth defect or reduced mtDNA copy number in the case of mtDNA$^{\Delta cob}$, the simulations predicted equal numbers of mtDNA$^{ATP6-NG}$ and mtDNA$^{\Delta cob}$ cells for the mtDNA$^{ATP6-NG}$ ×mtDNA$^{\Delta cob}$ cross, which was substantially different from the results of the respective FAST analysis, where mtDNA$^{\Delta cob}$ cells amounted to less than 20% (Fig 2F). Simulations for the mtDNA$^{ATP6-NG}$ ×mtDNA$^{\Delta cox2}$ cross, predicted reduced amounts (39.5%) of mtDNA$^{\Delta cox2}$ cells after 20 h of growth. However, this amount was still larger compared to the FAST analysis (27.7%). Thus, growth differences and unequal mtDNA contributions alone do not appear to be responsible for the experimentally observed depletion of mtDNA$^{\Delta cob}$ or mtDNA$^{\Delta cox2}$ cells in the respective populations. To further test this, we systematically varied the starting

ratios of mtDNA$^{ATP6-NG}$ to mutant mtDNA to identify conditions under which the simulations would reproduce the FAST results. The ratios required to match the experimental data were considerably lower—approximately 1:0.25 for mtDNA$^{\Delta cob}$ and 1:0.4 for mtDNA$^{\Delta cox2}$ —than the empirically measured starting amounts (Fig 2G).

In contrast to crosses involving mtDNA$^{\Delta cob}$ or mtDNA$^{\Delta cox2}$, for the mtDNA$^{ATP6-NG}$ ×mtDNA$^{\Delta atp6}$ cross, the simulated proportion of dark cells was only slightly higher (33.3%) than experimentally observed by FAST (30.8%), indicating that growth defects associated with mtDNA$^{\Delta atp6}$ cells in combination with reduced amounts of mtDNA$^{\Delta atp6}$ can account to a large extent for the reduction of mtDNA$^{\Delta atp6}$ cells in populations arising from cells heteroplasmic for mtDNA$^{ATP6-NG}$ and mtDNA$^{\Delta atp6}$.

In summary, our analyses indicate that segregation outcomes in populations derived from heteroplasmic zygotes are in part shaped by the combined effects of growth differences linked to mutant mtDNA and parental mtDNA copy-number disparities. These factors, which are explicitly incorporated into our simulations, are sufficient to explain the behaviour of mtDNA$^{\Delta atp6}$, whose depletion in FAST assays can be accounted for by its growth disadvantage together with its reduced mtDNA abundance in the parental haploid cell. In contrast, the experimentally observed loss of mtDNA$^{\Delta cob}$ — and, to a lesser extent, mtDNA$^{\Delta cox2}$ —is stronger than predicted by simulations that include only growth and mtDNA copy number effects. This discrepancy indicates the presence of an additional intracellular mode of selection against mutant mtDNA.

As shown previously, mitochondrial ultrastructure plays an important role in maintaining purifying selection [10]. It has been proposed that cristae-mediated compartmentalization of mitochondria allows each mitochondrial genome to supply proteins specifically to the respiratory chain complexes within its immediate vicinity, thereby enabling intracellular processes to selectively maintain intact genomes based on physiological differences in the mitochondrial network [10,11].

In our previous work, we demonstrated that deletion of *ATP20*, a gene required for proper cristae formation, compromises this spatial organization and impairs purifying selection against mtDNA lacking the *COB* gene [10]. To confirm these findings, to test our system, and to examine whether cristae formation is also critical for selection against mtDNA$^{\Delta cox2}$ and mtDNA$^{\Delta atp6}$, we performed FAST with strains lacking *ATP20*. Indeed, purifying selection was strongly reduced in Δ*atp20* cells and segregation shifted towards similar percentages of mtDNA$^{ATP6-NG}$ cells and cells containing mtDNA$^{\Delta cob}$, mtDNA$^{\Delta cox2}$ and mtDNA$^{\Delta atp6}$, implying that normal cristae formation via assembly of the dimeric ATP synthase [28] is critical for counterselection against mutant mtDNA (Fig 2D). The loss of counterselection against mtDNA$^{\Delta atp6}$ in the absence of Δ*atp20* was unexpected, as our analyses indicate that the disappearance of mtDNA$^{\Delta atp6}$ mtDNA in the FAST assays predominantly depends on growth defects and altered mtDNA copy number and should therefore be unaffected by the loss of *ATP20*, which has been implicated in intracellular selection against mutant mtDNA. We therefore examined the growth of mtDNA$^{\Delta atp6}$ strains in the Δ*atp20* background and found that the growth defect of mtDNA$^{\Delta atp6}$ cells was fully rescued, yielding growth indistinguishable from that of mtDNA$^{ATP6-NG}$ cells (S2H Fig). Moreover, mtDNA copy numbers in Δ*atp20* mtDNA$^{\Delta atp6}$ cells were not reduced compared to the Δ*atp20* mtDNA$^{ATP6-NG}$ strain, but rather increased. These findings explain the absence of mtDNA$^{\Delta atp6}$ depletion in the FAST assay, although the molecular basis for the suppression of growth defects and increase of mtDNA copy number by absence of Atp20 remain unclear.

Importantly, mtDNA copy numbers and growth rates were unchanged in Δ*atp20* strains containing mtDNA$^{ic}$ or mtDNA$^{\Delta cox2}$ compared with the corresponding WT strains containing mtDNA$^{ic}$ or mtDNA$^{\Delta cox2}$ (S2I Fig). This indicates that the counterselection against mtDNA$^{\Delta cox2}$ observed in FAST assays in a WT background is independent of mtDNA copy number differences and requires mechanisms that are absent in Δ*atp20* cells. Unexpectedly, mtDNA copy numbers increased substantially in Δ*atp20* strains carrying either mtDNA$^{il}$ or mtDNA$^{\Delta cob}$ relative to WT strains harboring the same mtDNA variants, a change that is likely responsible for the elevated proportion of dark cells lacking mtDNA$^{ATP6-NG}$ observed in the FAST analysis of mtDNA$^{ATP6-NG}$ ×mtDNA$^{il}$ and mtDNA$^{ATP6-NG}$ ×mtDNA$^{\Delta cob}$ crosses in a Δ*atp20* mutant background. The basis for this increase in intron-less mtDNA copy number upon loss of Atp20 remains unclear. Crucially, however, the relative ratio between mtDNA$^{il}$ and mtDNA$^{\Delta cob}$ is only mildly altered in a Δ*atp20* background (1:1) compared

with WT (1:0.8). Although the overall increase in mtDNA abundance may influence segregation dynamics to some extent, we consider it unlikely to fully explain the complete loss of counterselection against mtDNA$^{\Delta cob}$.

## Local differences of $\Delta\Psi$ and ATP levels within mitochondria of heteroplasmic zygotes

The sphere-of-influence hypothesis posits that each mitochondrial genome primarily supplies its immediate surroundings with gene products and thereby creates physiologically distinct mitochondrial domains that reflect the quality of the underlying mtDNA [10,11]. Although mitochondria form a continuous network with content equilibration of soluble matrix proteins, mtDNA and their membrane-bound gene products remain spatially constrained and do not mix rapidly [10]. Thus, mtDNA deletions could locally impair respiration, leading to reduced ATP production and lower $\Delta\Psi$ in their vicinity, facilitating their downstream removal. To test this hypothesis, we next asked whether mitochondrial physiology indeed differs between regions of the network that contain intact versus mutant mtDNA.

As a prerequisite to testing whether local physiological differences reflect mtDNA quality, we first used fluorescence microscopy to assess whether differences in $\Delta\Psi$ and ATP levels are detectable at the single-cell level in strains containing exclusively WT or mutant mtDNA. Relative ATP levels were quantified using the mitochondrially targeted ratiometric sensor mtQUEEN-2m, which reports ATP concentration based on shifts in its excitation spectrum upon ATP binding [29,30]. $\Delta\Psi$ was assessed using tetramethylrhodamine methyl ester (TMRM), a positively charged, lipophilic dye that accumulates in mitochondria in a $\Delta\Psi$-dependent manner [31,32]. To correct for mitochondrial content, TMRM fluorescence was normalized to the NG signal which was fused to the mitochondrial targeting sequence of subunit 9 of the *Neurospora crassa* ATP synthase (mtNG). This marker is known to localize to mitochondria in a manner largely independent of $\Delta\Psi$ [33]. First, we verified that mtQUEEN-2m localizes correctly to mitochondria. To this end, we compared mtQUEEN-2m fluorescence to that of mitochondrially targeted mKate2, which revealed a colocalization of both signals (S3A and S3B Fig). Next, we assessed whether expression of mtQUEEN-2m perturbs mitochondrial physiology. Neither cellular growth on YPG medium nor mitochondrial morphology-quantified in terms of network length-was detectably altered in the presence of mtQUEEN-2m (S3C and S3D Fig). WT cells exhibited significantly higher mitochondrial ATP levels and normalized mitochondrial TMRM signal compared to mtDNA$^{\Delta cob}$, mtDNA$^{\Delta cox2}$ and mtDNA$^{\Delta atp6}$ strains (Fig 3A–3D). These reductions resembled those observed in cells treated with carbonyl cyanide 3-chlorophenylhydrazone (CCCP), a protonophore that dissipates $\Delta\Psi$, or 2-deoxy-D-glucose (2-DG), which inhibits glycolysis and limits pyruvate supply to the TCA cycle, thereby reducing mitochondrial as well as cytosolic ATP levels to below 1% [30,34]. Notably, ATP levels measured with cytosolic cytQUEEN-2m, did not, or only slightly differ among all strains. mtDNA$^{\Delta atp6}$ strains showed slightly higher and mtDNA$^{\Delta cob}$ slightly lower levels of cytosolic ATP compared to their WT controls (S3E Fig). These findings indicate that the mutant strains exhibit severely compromised mitochondrial function. As expected, mtDNA$^{\Delta atp6}$ cells also showed a marked reduction in mitochondrial ATP levels relative to WT. Notably, $\Delta\Psi$ was also significantly reduced in mtDNA$^{\Delta atp6}$ cells, despite the presence of all genes encoding the structural components necessary for the assembly of complexes III and IV, which in principle should be capable of generating a $\Delta\Psi$. This finding aligns with previous reports demonstrating that the absence of an assembled ATP synthase can impair complex IV stability, thereby compromising the respiratory chain's ability to maintain $\Delta\Psi$ [18,19].

To explore whether physiological differences between mitochondrial segments containing intact or mutant mtDNA persist locally within fused mitochondrial networks, we performed mating experiments to generate heteroplasmic zygotes and assessed local physiological differences (Fig 3E). To this end, we imaged mtQUEEN-2m ratios and TMRM intensities in zygotes formed by mating WT cells with cells containing either WT or mutant mtDNA. Importantly, mitochondria fuse in zygotes and matrix components equilibrate, while mtDNA and mtDNA encoded subunits show limited diffusion [10] (S4A and S4B Fig). To distinguish the mating partners in microscopy images, the cell walls of cells containing intact mtDNA (P1) were labeled with concanavalin A (conA) linked to the far-red dye Alexa 633. Notably, in these mating assays, the P1 cell carried intact mtDNA but not the version encoding the Atp6–mNeonGreen fusion, thereby preventing

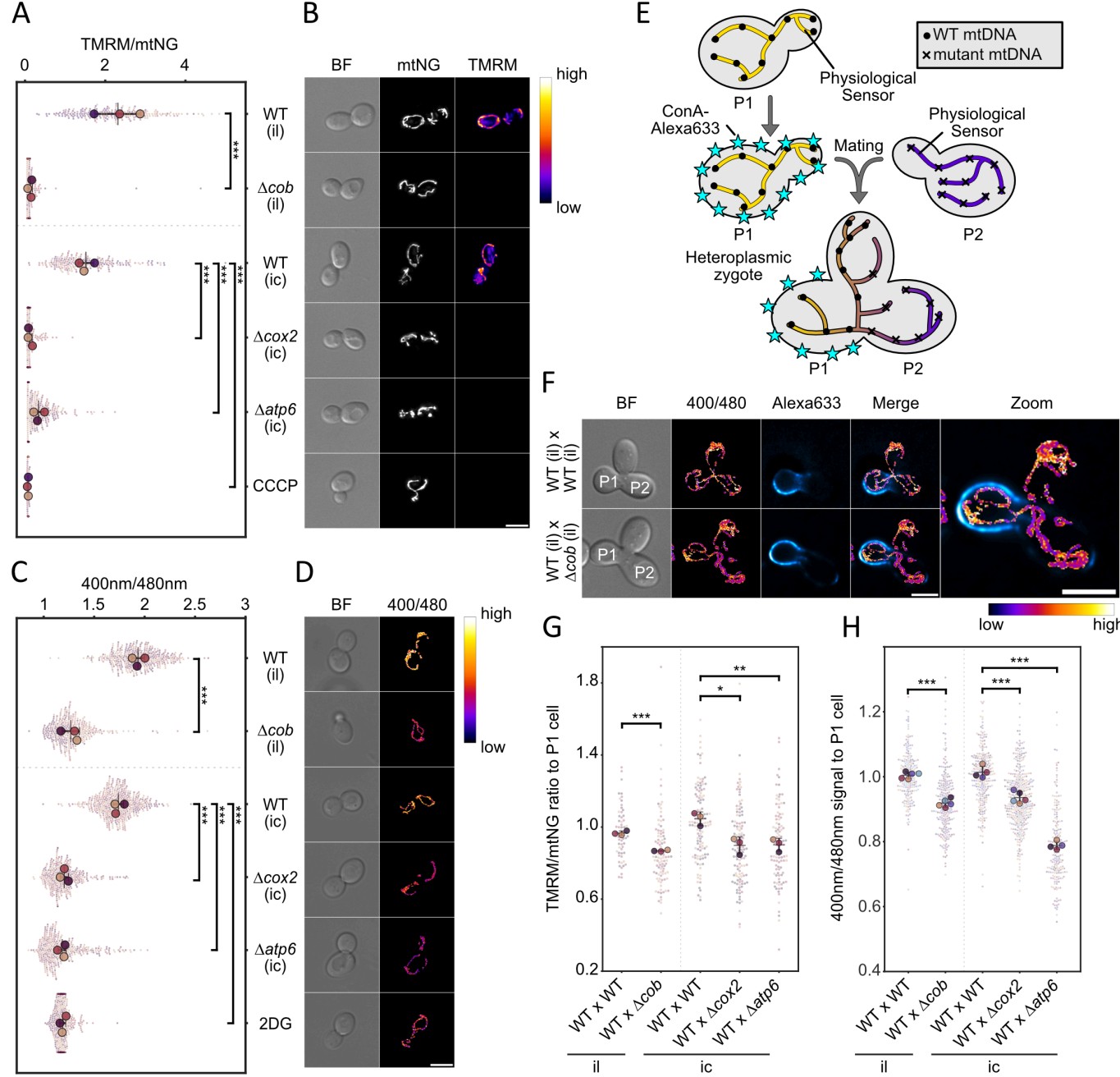

**Fig 3. Local differences of ΔΨ and ATP levels within mitochondria of heteroplasmic zygotes. (A)** Superplot of membrane-potential measurements in WT and mutant mtDNA strains; large dots denote biological-replicate means, and smaller dots represent single-cell values. **(B)** Maximum intensity projections of widefield microscopy images of cells containing indicated mtDNAs stained with 50 nM TMRM. Nuclear-encoded Su9-NG (mtNG) was used as a mitochondrial marker. Data in (A) are expressed as TMRM/mtNG intensity ratios, reflecting ΔΨ normalized to mitochondrial mass. WT cells treated with 10μM CCCP serve as a depolarized control. **(C)** Quantification of relative ATP levels in strains expressing mitochondrially targeted mtQUEEN-2m, obtained by determining fluorescence after excitation with 400 nm and 480 nm. WT cells in synthetic medium containing 2-deoxy-glucose instead of glucose provide an ATP-depleted control. **(D)** Representative images of mtQUEEN-2m–expressing cells, with pixel-wise 400 nm/480 nm ratio computed using the Image Calculator in FIJI. **(E)** Experimental outline of the mating of two haploid cells to generate a heterozygous zygote with a labelled P1 cell. **(F)** Representative widefield fluorescence images of zygotes expressing mtQUEEN-2m; parental cells share identical mtDNA in the upper row, or comprise WT and mutant mtDNA in the lower row. Bright-field panels indicate P1 and P2 cells. Scale bar for all images ((B), (D) and (F)) = 5μM. **(G, H)** Quantification of ΔΨ **(G)** and ATP levels **(H)** in zygotes, expressed as the ratio of values measured in the P2 cell relative to those in the P1 cell. Panels (A), (C), (G), and (H) were analysed for statistical significance by unpaired Student's t-test on replicate means (*P <0.05; **P <0.005; ***P < 0.001).

misinterpretation of the signals originating from the fluorescent sensors. In zygotes formed by matings between cells both containing intact mtDNA, the fused mitochondrial network cells exhibited similar ATP levels and $\Delta\Psi$ within the segments that were present within the P1 cell and the mating partner (P2) (Fig 3F–3H and S4C Fig). In contrast, zygotes obtained from matings between cells containing intact mtDNA (P1) with cells containing mutant mtDNA (P2) retained marked differences. ATP levels and $\Delta\Psi$ remained higher in the WT P1 cell compared to the mutant P2 parent cell. Although the mitochondrial network in the P2 parent cell exhibited increased ATP and $\Delta\Psi$ relative to their haploid forms, likely reflecting some degree of metabolite exchange, mutant-derived compartments did not reach WT-like levels.

These observations support the idea that mitochondrial physiological parameters, such as ATP levels and $\Delta\Psi$, are locally influenced by the underlying mtDNA genotype and are not fully equilibrated within the fused network. This provides a functional basis for the sphere-of-influence model and suggests that local differences in physiology may enable the selective recognition of defective mitochondrial genomes.

## Functional respiratory complexes are required for selective mtDNA inheritance

We reasoned that if these local physiological differences underlie mtDNA quality control, then impairing the respiratory chain globally should abolish selection against mutant mtDNA.

According to the idea that $\Delta\Psi$ or ATP levels generated by the OXPHOS complexes serve as signals to distinguish intact from mutant mtDNA, a general impairment of the electron transport chain in heteroplasmic cells should reduce and equalize $\Delta\Psi$ and ATP levels across the entire mitochondrial network. This would obscure differences in mtDNA quality and thereby impair the selective detection of intact versus mutant genomes.

To test this hypothesis, we generated strains with nuclear deletions of genes encoding subunits from either complex III or complex IV by deleting *RIP1* or *COX4*, respectively. We also deleted *ATP4* to examine the effects of impaired ATP synthase function. However, these *ATP4* deletion strains exhibited pronounced mtDNA instability, in line with previous reports [35], which precluded the analysis of purifying selection in this background. First, we examined ATP levels and $\Delta\Psi$ of $\Delta cox4$ and $\Delta rip1$ strains containing intact or mutant mtDNA. As expected, ATP levels and $\Delta\Psi$ in these strains dropped to levels observed also in strains containing mutant mtDNA in a WT background (S4E and S4F Fig). Notably, mating between cells containing intact mtDNA with cells containing mtDNA$^{ic}$, mtDNA$^{il}$, mtDNA$^{\Delta cob}$, mtDNA$^{\Delta cox2}$ or mtDNA$^{\Delta atp6}$ in $\Delta rip1$ or $\Delta cox4$ backgrounds, we found that heterogeneities in the fused mitochondria of the resultant zygotes were no longer detectable (Fig 4A–4D).

We used flow cytometry to assess whether respiratory-deficient mutants continued to express Atp6-NG, a prerequisite for analysing mtDNA quality control during FAST. Fluorescece was observed in these cells, however, the proportion of non-fluorescent cells modestly increased in $\Delta rip1$ (7.4%) and $\Delta cox4$ (6.7%) strains compared to WT (2.0%) (S5A Fig). We then performed FAST analyses to investigate the segregation of intact and mutant mtDNA in the progeny of heteroplasmic zygotes lacking *RIP1* or *COX4*. Notably, by using mtDNA$^{ATP6-NG}$, the FAST assay enabled us to directly detect the presence of intact mtDNA even in respiratory-deficient $\Delta rip1$ or $\Delta cox4$ mutants, which is an advantage not afforded by our previous pedigree analyses, which relied on respiratory growth as a proxy for mtDNA integrity [10]. In the FAST mating experiments, purifying selection against mtDNA$^{\Delta cob}$ or mtDNA$^{\Delta cox2}$ was abolished in $\Delta rip1$ and $\Delta cox4$ backgrounds, resulting in an approximately equal distribution of cells expressing green fluorescence and those lacking fluorescence (Fig 4E).

To further dissect why purifying selection against mtDNA$^{\Delta cob}$ and mtDNA$^{\Delta cox2}$ is lost in $\Delta rip1$ and $\Delta cox4$ backgrounds, we first examined mtDNA copy number and growth of these mutants carrying either intact or mutant mtDNA. Compared to WT cells, where both mtDNA$^{\Delta cob}$ and mtDNA$^{\Delta cox2}$ show a modest reduction in copy number relative to their intact counterparts, $\Delta rip1$ and $\Delta cox4$ mutants displayed uniformly lower mtDNA abundance for both intact and mutant genomes, and the modest WT decrease of mutant mtDNA compared to the respective intact mtDNA was no longer detectable (S5D Fig). Notably, mtDNA$^{\Delta cob}$ copy numbers were even slightly elevated in a $\Delta cox4$ strain compared to the mtDNA$^{il}$ in

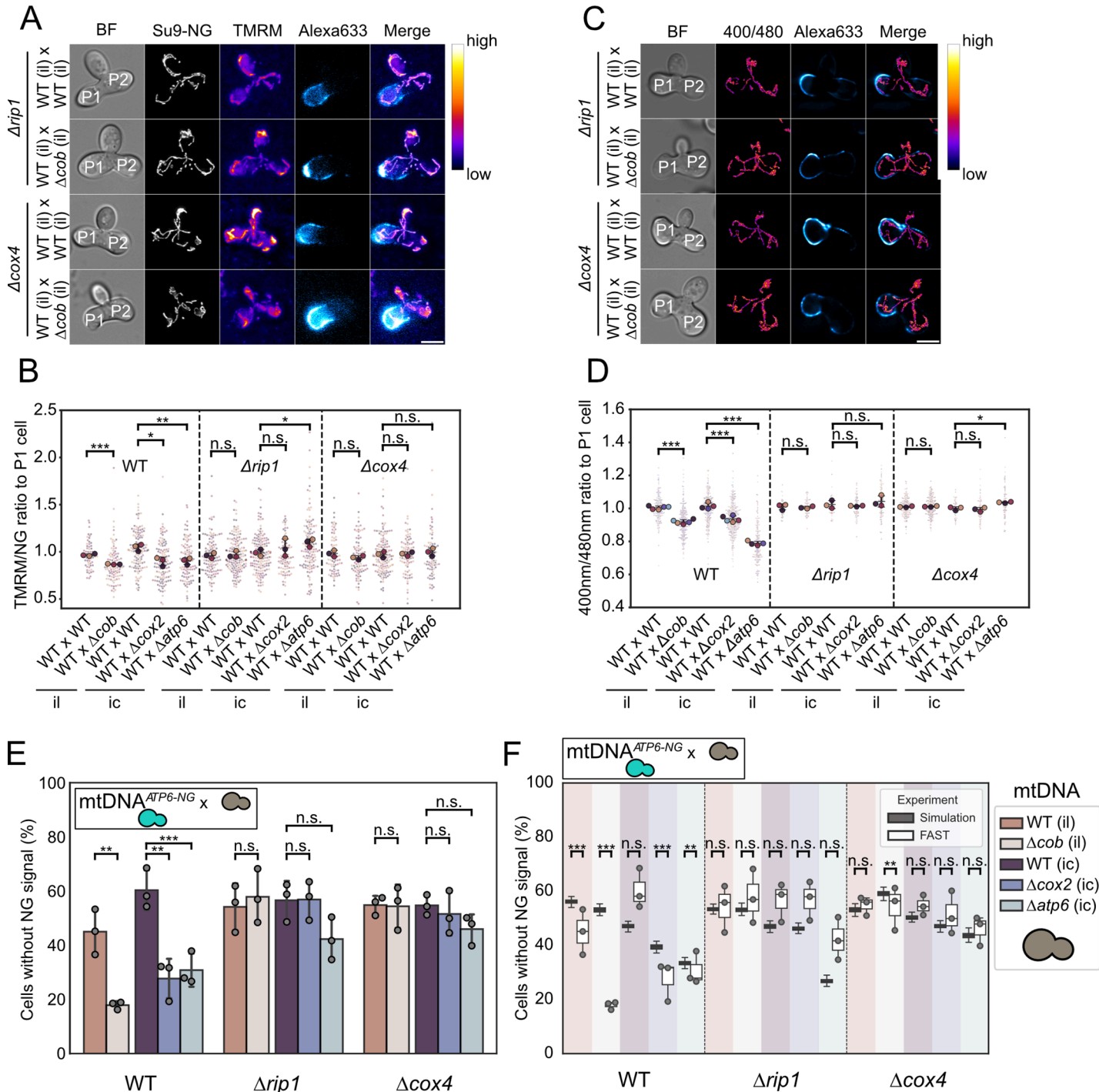

**Fig 4. Functional Respiratory Complexes Are Required for Selective mtDNA Inheritance. (A)** Maximum-intensity projections of widefield fluorescence images showing ΔΨ in heteroplasmic zygotes of Δ*rip1* and Δ*cox4* strains. **(B)** Superplot of P2 to P1 ΔΨ ratios in heteroplasmic Zygotes. **(C)** Maximum-intensity projections depicting mitochondrial ATP levels in heteroplasmic zygotes of Δ*rip1* and Δ*cox4* strains. **(D)** Superplot of P2 to P1 mitochondrial ATP level ratios in heteroplasmic zygotes. **(E)** FAST analysis results for WT Δ*rip1* and Δ*cox4* matings. **(F)** Simulated vs experimental outcomes of FAST analysis for WT, Δ*rip1* and Δ*cox4* matings. Panels (B), (D) and (E) were analysed for statistical significance by unpaired Student's t-test on replicate means. Statistical significance in panel (F) was determined by Monte Carlo comparison of each FAST replicate to the simulation distribution, with replicate p-values combined using Fisher's method. (*P <0.05; **P <0.005; ***P < 0.001).

$\Delta cox4$. Furthermore, post-growth mixing assays revealed no evidence for altered competitive fitness between $\Delta rip1$ or $\Delta cox4$ cells carrying intact versus mutant mtDNA (S5B Fig). Unlike in WT backgrounds—where our simulations failed to reproduce the experimental FAST outcomes—simulations for $\Delta rip1$ and $\Delta cox4$ backgrounds matched the measured segregation patterns or underestimated the percentage of dark cells. Taken together, these results suggest that when respiration is abolished by nuclear mutations, intracellular selection against mtDNA$^{\Delta cob}$ or mtDNA$^{\Delta cox2}$ mtDNA is lost. Since only minor differences in mtDNA copy numbers and growth exist between strains containing mtDNA$^{\Delta cob}$ or mtDNA$^{\Delta cox2}$ and their respective controls, also purifying selection against mtDNA$^{\Delta cob}$ and mtDNA$^{\Delta cox2}$ in populations derived from heteroplasmic zygotes is entirely lost.

Our FAST analysis using WT nuclear background combined with simulations did not suggest a strong influence of intracellular selection on the purifying selection against mtDNA$^{\Delta atp6}$. Nevertheless, we performed FAST analyses examining the competition between mtDNA$^{\Delta atp6}$ and mtDNA$^{ATP6\text{-}NG}$ in $\Delta rip1$ and $\Delta cox4$ backgrounds. This analysis revealed a slightly lower proportion of non-fluorescent cells lacking mtDNA$^{ATP6\text{-}NG}$ in $\Delta rip1$ or $\Delta cox4$ backgrounds (Fig 4E). However, $\Delta rip1$ cells, and to a lesser extent $\Delta cox4$ cells, harboring mtDNA$^{\Delta atp6}$ showed growth defects compared to the corresponding strains carrying intact mtDNA$^{ATP6\text{-}NG}$ (S5B and S5C Fig). This likely accounts for the underrepresentation of non-fluorescent cells in the mating experiments, as these cells are outcompeted by haploid progeny carrying the intact mtDNA$^{ATP6\text{-}NG}$. The growth defects of $\Delta rip1$ and $\Delta cox4$ strains harboring mtDNA$^{\Delta atp6}$ likely stems from their inability to establish a $\Delta\Psi$—neither via electron transport chain activity nor through reverse operation of the ATP synthase. Comparison of the FAST results with simulations that account only for growth defects and mtDNA copy numbers—both assessing matings between mtDNA$^{ATP6\text{-}NG}$ and mtDNA$^{\Delta atp6}$ in a $\Delta rip1$ or $\Delta cox4$ background—showed that the fraction of dark cells in the FAST assay was not lower than in the simulations (Fig 4F). This is consistent with the conclusion that counterselection against mtDNA$^{\Delta atp6}$ is absent in these mutants.

## Discussion

In this study, we introduce FAST, a novel pipeline to analyse the segregation behaviour of mtDNA variants in heteroplasmic *S. cerevisiae* zygotes. Using this approach, we demonstrate purifying selection against mutant mtDNA genomes lacking components of complexes III, IV, or the ATP synthase in populations arising from heteroplasmic founder cells. This purifying selection reflects the combined contribution of the following processes: (i) the relative abundance of intact and mutant genomes present in the founding zygote, (ii) growth defects associated with cells carrying mutant mtDNA, and (iii) an additional mechanism that cannot be explained by these factors alone and likely represents an intracellular disadvantage of the mutant genome. Selection against mtDNA$^{\Delta cob}$, mtDNA$^{\Delta cox2}$, and mtDNA$^{\Delta atp6}$ relies on these processes to different extents. For mtDNA$^{\Delta cob}$ and, to a lesser degree, mtDNA$^{\Delta cox2}$, our data indicate a intracellular component, whereas intracellular selection against mtDNA$^{\Delta atp6}$ is minimal.

The weak or potentially even entirely absent intracellular selection against mtDNA$^{\Delta atp6}$ is reminiscent of findings in mammalian systems, where weaker selection against mutation in the *ATP6* gene was observed [9,36–38]. We propose that this may be linked to the structural role of dimeric ATP synthase in shaping cristae morphology. Loss of *ATP6* disrupts dimer formation of the ATP synthase and compromises cristae organization [18], which may impair the spatial segregation of mitochondrial gene products, the so-called "sphere of influence". Increased mixing of gene products between mitochondria harboring functional and mutant genomes could blur the physiological differences needed to drive selective maintenance of intact mtDNA.

Although both *COB* and *COX2* encode essential subunits of the electron transport chain, there appears to be substantially stronger intracellular selection against mtDNA$^{\Delta cob}$ than mtDNA$^{\Delta cox2}$. We note that mtDNA$^{\Delta cob}$ was assessed in an intron-less background, whereas mtDNA$^{\Delta cox2}$ was examined in an intron-containing mtDNA context. It remains to be determined to what extent this difference contributes quantitative effects on mtDNA maintenance or expression. Nonetheless, the magnitude of the difference suggests that the nature of the respiratory defect plays an important role in

shaping intracellular selection of mtDNA. Complex III and IV act at different positions within the electron transport chain, and blocking electron flow at distinct steps may have different physiological consequences. Thus, loss of Cob and loss of Cox2, although both disrupting respiration, may not be equivalent in how they affect mitochondrial redox state and overall organelle function (e.g. [39]). Moreover, defects in the assembly of Complex III and Complex IV are not equivalent, and differences in the types or stability of assembly intermediates due to the absence of *COB* or *COX2*, respectively, could lead to distinct physiological consequences [40,41]. Such differences may cause the stronger selection against mtDNA$^{\Delta cob}$. Together, our findings suggest that the specific consequences associated with different mutant mtDNA variants can shape the underlying selection processes and thereby modulate intracellular selection strength. Along similar lines, it will be important to determine whether mutations affecting other classes of mtDNA-encoded genes, such as tRNAs or rRNAs, which are common lesions in human mitochondrial disease, elicit distinct selection behaviors. FAST now offers the opportunity to systematically examine how a broad spectrum of mtDNA mutations influences heteroplasmic dynamics in populations.

Our findings reveal that mtDNA selection depends on a functional respiratory chain: in cells lacking nuclear-encoded components such as *COX4* or *RIP1*, selection against mutant mtDNA is abolished. These findings support the sphere of influence model in which mitochondrial function, likely through outputs such as ATP production or $\Delta\Psi$, drives local mtDNA selection. While the mitochondrial network is continuous, our data demonstrate that regions can maintain distinct physiological profiles depending on the quality of a nearby mtDNA copy. Indeed, we detect spatial differences in $\Delta\Psi$ and ATP levels within fused zygotic mitochondria, and propose that similar heterogeneity exists in other heteroplasmic contexts. Although such compartmentalization seems at odds with mitochondrial fusion and diffusion, prior studies have reported similar local variation in $\Delta\Psi$ [42–45], but without linking it to mtDNA content.

Interestingly, zygotes arising from crosses between cells containing intact mtDNA and mtDNA$^{\Delta atp6}$ still display physiological heterogeneity. Regions derived from the mtDNA$^{\Delta atp6}$-containing parent show reduced ATP levels and lower membrane potential relative to regions populated by intact mtDNA. Although this may appear inconsistent with the absence of intracellular selection against mtDNA$^{\Delta atp6}$, a possible explanation is that pre-existing cristae in the parental compartment containing intact mtDNA are preserved in these matings. Because mtDNA$^{\Delta atp6}$ does not impair the function of already-assembled ATP synthase dimers and cristae, respiratory capacity in these regions remains largely intact. In contrast, daughter cells arising from these zygotes must build new mitochondrial architecture *de novo*, including cristae and respiratory chain complexes. We propose that, under these conditions, the establishment of locally distinct mitochondrial subdomains with differential physiological performance is diminished, thereby preventing effective intracellular counterselection against mtDNA$^{\Delta atp6}$.

A major unresolved question is how mitochondrial physiology is translated into differential maintenance of mtDNA. One possibility is that local reductions in ATP or $\Delta\Psi$ impair the import of nuclear-encoded proteins required for mtDNA replication or maintenance, thereby preventing propagation of defective genomes. Such a mechanism has been proposed in *Drosophila*, where local depolarization prevents import of replication factors [5,7,46]. Beyond direct effects on import, reduced $\Delta\Psi$ may also trigger signaling pathways, for example, via phosphorylation of the RNA-binding protein LARP by the kinase Pink1 [7], which inhibit local cytosolic protein translation on mitochondria. However, this pathway is absent in *S. cerevisiae*, which lacks both Pink1 and LARP. This suggests that yeast cells may employ analogous but mechanistically distinct strategies to couple mitochondrial function to mtDNA quality control, likely involving yet unidentified factors.

Alternatively, selection may be enforced by differential transport of mitochondrial fragments during budding. The adaptor protein Mmr1, which links mitochondria to Myo2, has been implicated in directing functional mitochondria to daughter cells [47–49]. Spatial segregation of functional and non-functional mitochondria based on their physiological output could reinforce selective inheritance during cell division. However, the molecular mechanisms that allow Mmr1 to distinguish mitochondrial quality remain to be determined.

Recent work in mammalian systems has shown that population-level shifts in heteroplasmy can arise primarily from selection acting on cell fitness [4], yet the relative contributions of founder-cell mtDNA stoichiometry, growth-associated

fitness defects, and potential intracellular selection remain difficult to disentangle in these contexts. Our study establishes *S. cerevisiae* as a system in which these layers can be resolved with precision. Through quantitative integration of mtDNA copy-number measurements, growth analyses, and the FAST assay, we distinguish effects driven by initial mtDNA ratios in heteroplasmic founder cells, from population-level growth disadvantages of cells harboring mutant genomes, and from a third, genetically separable intracellular mode of selection that specifically disfavors certain mutant mtDNA variants. Taken together, our results underscore the importance of mitochondrial function in guiding mtDNA selection and establish *S. cerevisiae* as a powerful system to dissect the underlying mechanisms. The ability to model heteroplasmy, perform genetic manipulation, and apply scalable assays such as FAST opens the door to systematic dissection of mitochondrial quality control across diverse classes of mtDNA mutations, and provides a conceptual and methodological foundation for future studies aimed at resolving how distinct selective forces shape heteroplasmy dynamics.

## Materials and methods

### Yeast strains and plasmids

All yeast strains are derived from the W303 background. Deletion of genes was performed using homologous recombination according to protocols described in [50]. Strains generated or used for this study can be found in S1 Table. The plasmids used are listed in S2 Table.

### Yeast growth conditions

Yeast strains were kept on either rich media (1% yeast extract, 2% peptone, 0.04% adenine, 2% bacto-agar) with 3% glycerol (WT mtDNA/functional respiratory chain) or synthetic media lacking arginine (0.67% yeast nitrogen base, 2% glucose, 0.192% drop-out mix minus arginine) (mtDNA$^{\Delta cob}$, mtDNA$^{\Delta cox2}$ and mtDNA$^{\Delta atp6}$) to ensure mtDNA maintenance prior to experiments. Unless stated otherwise, cells were grown overnight in rich media containing 2% glucose (YPD) at 30°C with 170 rpm shaking. In the morning, cultures were diluted to an $OD_{600}$ of 0.1 and grown into early to mid log phase ($OD_{600}$ 0.6-0.8).

### Fast pedigree analysis (FAST)

For mating, 500 µL of each strain in mid-log phase ($OD_{600}$ 0.6–0.8) and of opposing mating type were mixed and transferred onto an YPD plate. Mating was conducted for 2 hours at 30°C. Following incubation, a tiny amount of cells was gently scraped from the plate using a pipet tip, resuspended in 30 µl of ddH$_2$O, and pipetted onto a new YPD plate. The plate was tilted to allow the droplet to form a line, creating an optimal cell density for microdissection. The plate was then incubated at 30°C for an additional 30 minutes to promote sufficient growth of the first daughter cell from the zygote, allowing it to be detached from the zygote. After this period, 30 daughter cells were selected and relocated using a microdissection microscope (Singer Sporeplay+). Cells were allowed to grow into colonies for 20 h at 30°C. For flow cytometry, the formed colony was scraped off the plate with a pipette tip and resuspended in 300 µl sterile filtered SC medium. The samples were analysed with a BD Accuri C6 Plus Flow Cytometer with a limit of 10,000 events per biological replicate. Initial analysis and gating was performed using the floreada.io tool followed by custom python scripts. To exclude false hits (e.g. dirt, agar) the events were gated for FSC-A (Forward-Scatter - Area) vs FSC-H (Forward-Scatter -Height).

### Cell growth determination

To determine cell growth, overnight cultures were diluted into fresh medium and incubated at 30 °C until reaching mid logarithmic phase ($OD_{600} \sim$ 0.6-0.8). Cells were harvested by centrifugation, washed three times with sterile deionized water, and resuspended in YPD medium to an $OD_{600}$ of 0.1. Cells were transferred into the wells of a 96-well plate, and growth was recorded at 30 °C on a SPECTROstar Nano (BMG Labtech) plate-reader at 20-minute intervals for up to

24 h, with continuous shaking at 800 rpm. Each strain was assayed in three technical replicates per plate. $OD_{600}$ measurements were logarithmically transformed and fitted by linear regression to obtain the exponential growth rate (r, slope of the regression); doubling time ($T_d$) was then calculated as

$$T_d = \frac{\ln(2)}{r}$$

### Genomic DNA isolation and quantitative PCR

Mid-log phase cells corresponding to $3.75 \cdot 10^7$ cells were harvested from back-diluted overnight cultures by centrifugation (3000 g, 3 min at RT). Pellets were washed with sterile deionized water and stored at -80°C. Genomic DNA (gDNA) was extracted via bead-beating phenol-chloroform extraction protocol as described previously [51]. DNA concentration was quantified using a NanoPhotometer (Implen N60-Touch), and all samples were diluted to a final concentration of 0.1 ng/μl. Quantitative PCR was performed using primers targeting *ACT1* and *COX1*, following the methodology outlined in [52]. mtDNA abundance was determined by calculating the ratio of *COX1* to *ACT1*. Relative abundance was then derived by comparing the mean biological replicates of the sample strains to those of the reference strain.

### Mating and stainings for microscopy

To distinguish P1 and P2 cells in zygotes during microscopy, WT cells (P1) were stained with conA-Alexa633 prior to mating. 1 mL of log phase cells (0.6 - 0.8) were spun down at 3000 g for 3 minutes. After decanting the supernatant, the cells were incubated at 30°C for 15 minutes in 500 μL 1x PBS containing 10 μg/mL conA-Alexa633 (ThermoFisher). Then, to mate the strains, 700 μL of each strain in mid-log phase were mixed in a reaction tube and centrifuged at 3000 g for 3 minutes. The supernatant was discarded and the cell pellet was resuspended in 50 μL of YPD medium. The suspension was applied to a YPD agar plate and gently spread into a line by tilting the plate. The liquid was then allowed to absorb/dry for 5–10 minutes. Plates were subsequently incubated at 30°C. After 90 minutes cells were scraped off the plate, and resuspended in 2 mL YPD. The $OD_{660}$ was determined and $2 \cdot 10^6$ cells were taken, centrifuged at 3000 g for 3 minutes and the supernatant discarded. The cells were then incubated at 30°C for 10 minutes in 1 mL 10 mM HEPES pH 6.7 supplemented with 50 nM Image-iT TMRM reagent (ThermoFisher), using a tube rotor (IKA Trayster digital). Then, the cells were washed twice in 1x PBS and loaded on a conA coated Ibidi slide (Ibidi GmbH) for imaging. Identical TMRM staining protocols were used for zygotes and single cells. For the DAPI staining of zygotes shown in S4A and S4B Fig, conA-Alexa633 staining and mating were performed as described above. Subsequently, $OD_{660}$ was determined and $2 \cdot 10^6$ cells were collected by centrifugation at 3000 g for 3 minutes. The supernatant was discarded and the pellet was resuspended in 1 mL of SC medium supplemented with 2 μg/mL DAPI. Cells were incubated for 15 minutes at 30°C with gentle rotation using a tube rotor. After staining, cells were pelleted again, resuspended in 400 μL of SC medium, and transferred to a conA coated well of an Ibidi slide for imaging.

### Fluorescence microscopy

Imaging was performed with a Nikon Ti2-Eclipse wide-field fluorescence microscope with a CFI Apochromat TIRF 100×/1.49 numerical aperture (NA) oil objective and a Photometrics Prime 95B 25mm camera. The microscope is encased by an environmental box to keep the temperature at 30°C. Yeast cells were immobilized on the surface of Ibidi 8-well μ-Slides (18-well slides were used for pedigree imaging) using pretreated wells with conA (1 mg/mL). Z-stacks were acquired using a step size of 0.2 μm, spanning a total range of 8 μm. The ratiometric ATP sensor QUEEN-2m was targeted either to mitochondrial matrix (mtQUEEN-2m) by fusion to the mitochondrial presequence Su9, or to the cytosol (cytQUEEN-2m), which lacks a dedicated targeting sequence. QUEEN-2m was sequentially excited with 395 nm and 480 nm using SepctraX LED engine in triggered acquisition mode. For both excitation wavelengths, fluorescence emission around 520 nm was collected using the GFP filter cube (F47-525NXL, 525/50).

## Image processing and analysis

All microscopy images were deconvolved with the Huygens software (Scientific Volume Imaging). Analysis of the FAST experiments shown in S2A and S2B Fig has been performed as described previously [12]. Budding cells and zygotes in microscopy images were automatically identified and segmented using YeastMate [53] within custom FIJI macros, followed by cropping of individual cells into single images. Mitochondrial segmentation was performed in three dimensions using Mitograph, and channel intensities were summed along the network coordinates provided by Mitograph (dimensions: xy= 0.11, z=0.2) [54]. Zygotes were subdivided into P1, P2, and daughter cells through the use of binary masks. These masks were generated semi-automatically with a custom macro using the YeastMate plugin in FIJI. Channel signals within the zygotes were then assigned to the corresponding parental or daughter cells based on the masks. QUEEN fluorescence was analysed as follows: For mtQUEEN the 480 nm channel was segmented using MitoGraph, and voxel intensities along the resulting mitochondrial network traces were extracted. For each network point, the ratio between the fluorescence intensities obtained from 400 nm and 480 nm excitation was computed. cytQUEEN signals were segmented using Otsu thresholding, and within the resulting three-dimensional masks, ratiometric values were obtained by calculating the ratio of the summed intensities from both excitation channels. TMRM signals were summed up along the mitochondrial network obtained by MitoGraph segmentation of the mtNG channel and ratios to mtNG were calculated. Mitochondrial network length was computed from the MitoGraph vertex table by summing the Euclidean distances between consecutive 3D coordinates ($x,y,z$) along each mitochondrial line (line_id), and then adding these lengths across all lines in the cell.

## Simulations

For the simulations in Fig 2F and 2G, we adapted a previously established framework [12] by introducing a growth-rate adjustment and a different mtDNA copy number for the cells containing mutant mtDNA. First of all, we calculated the probability that, within a simulation step corresponding to the doubling time of the WT, a given cell divides and has a daughter cell $p_{daughter}$ as follows:

$$p_{daughter} = 2^{g_i} - 1$$

where $g_i$ is the growth rate ratio (compared to WT) for the cell at time point $i$, which is calculated as follows:

$$g_i = (g_{other} - 1)h_i$$

where $g_{other}$ is the growth rate penalty of the strain crossed with WT (0.99 for mtDNA$^{il}$, 1.03 for mtDNA$^{\Delta cob}$, 1.01 for mtDNA$^{ic}$, 0.98 for mtDNA$^{\Delta cox2}$ 0.93 for mtDNA$^{\Delta atp6}$, and 1.0 for WT), while $h_i$ is the mean of zeros and ones (representing the alleles of nucleoids) for the cell at time point $i$. Next, we ran the simulation 100 times for the experimentally measured mtDNA ratios and 10 times for simulated mtDNA ratios. For each run, we simulated 30 cells and calculated the average $h$. Note that all simulations were performed starting from a cell with 32 nucleoids distributed uniformly as sequence [1,0,1,0,1,0...] (see S2G Fig). To account for different mtDNA copy numbers between the crossed strains, ones and zeros in random positions along the sequence were flipped to achieve the nearest possible zeros-to-ones ratio compared to the required mtDNA ratio for that run (e.g., with mtDNA ratio = 0.1, the resulting sequence would contain 3 ones and 29 zeros, since 32*0.1=3.2 rounded to the nearest integer = 3). The other parameters of the simulation are *ngen*, *ndau*, and *nspl*. For the meaning of the parameters, we refer the reader to reference [12]. For Fig 2F and 2G, we used the following values: *ngen* = 14, *ndau* = 11, and *nspl* = 5. Given the calculated WT strain doubling time of 1.48 hours, the number of generations *ngen* = 14 was chosen to simulate at least 20 hours, as in the experiments. To assess whether the FAST measurements deviated significantly from the distribution predicted by our simulation model, we performed a

Monte-Carlo–based hypothesis test for each biological replicate. For a given strain, each FAST replicate value was compared against a null distribution generated by repeatedly sampling (with replacement) from the corresponding simulation values. For each replicate, an empirical one-sided p-value was calculated as the proportion of Monte Carlo samples that were less extreme than the observed FAST value, testing the hypothesis that the simulation outcome exceeds the FAST measurement. The three replicate-level p-values were then combined using Fisher's method to obtain a single strain-specific significance value.

## Supporting information

**S1 Table. Strain list.**
(DOCX)

**S2 Table. Plasmid list.**
(DOCX)

**S1 Fig. Proportion of heteroplasmic population in FAST using zygotes vs first daughters. (A)** Scatterplot of the data shown in Fig 1B, illustrating the minor fraction of dark cells. The population of green fluorescent cells is highlighted by the red circle. **(B)** KDE plot of FAST analysis of mtDNA$^{ATP6-NG}$ mated with mtDNA$^{ic}$ when the first daughter cell of the Zygote was picked. The heteroplasmic cell population is highlighted by the red circle. **(C)** Histogram of fluorescence intensity levels of mtDNA$^{ATP6-NG}$ cells with a *PDR5* deletion following 1mg/mL chloramphenicol (CAP) treatment. Atp6-NG fluorescence half-life was determined by fitting an exponential decay to the log-transformed fluorescence medians over time. **(D)** Histogram of the NG intensity distribution among FAST population after 20 h of mtDNA$^{ATP6-NG}$ mated with mtDNA$^{ic}$ when zygotes or daughters of the zygotes were picked. Dotted lines mark the two peak maxima and the interpeak valley; shaded regions denote population assignments. The heteroplasmic window is defined symmetrically around the valley with a width of ±10 % of the interpeak distance. **(E)** Schematic of the modified FAST workflow: instead of harvesting zygotes, first-division daughters are detached and relocated on agar. **(F)** Drop-dilution assay of mtDNA-variant strains on fermentable (glucose) and non-fermentable (glycerol) rich media. Plates were incubated at 30 °C and imaged after 48 h. **(G)** Crosses of mtDNA$^{\Delta cob}$, mtDNA$^{\Delta cox2}$, and mtDNA$^{\Delta atp6}$ strains against tester strains to screen for secondary deletions in the mitochondrial genome. Crosses were replicaplated on non-fermentable medium (YPG) to assess growth rescue. The right panels show the corresponding master plate of streaked cells on SC-Arg. White asterisks indicate colonies that failed to grow and were excluded from analysis; white arrows indicate crosses that did not rescue growth on YPG. **(H)** Representative images of DAPI stainings of mtDNA$^{\Delta cob}$, mtDNA$^{\Delta cox2}$, mtDNA$^{\Delta atp6}$, and $\rho^0$ cells expressing mtKate2 as a mitochondrial marker, used to visualize mtDNA. Note that no $\rho^0$ cells were identified in any of the strains (n>200). Panel (C) was analysed for statistical significance using an unpaired student T-test was performed on the means of the replicates (*P <0.05; **P <0.005; ***P < 0.001).
(TIFF)

**S2 Fig. Confirmation of FAST by microscopy and determination of cell growth and mtDNA copy number. (A)** Representative widefield fluorescence microscopy images of microcolonies grown for 20 h following isolation of 30 first-division daughters from mtDNA$^{ATP6-NG}$ ×mtDNA$^{ic}$, mtDNA$^{\Delta cob}$, mtDNA$^{il}$, mtDNA$^{\Delta cox2}$, or mtDNA$^{\Delta atp6}$ matings. **(B)** Microscopy-based FAST quantification: individual daughter cells were arrayed on distinct plate positions, enabling each to form a microcolony over 20 h; colonies were then relocated to separate wells for imaging. Approximately 200 cells per well were scored for fluorescence status, with the ratio of fluorescent to non-fluorescent cells plotted as individual points in a swarmplot. The three big dots represent the mean of a single replicate. **(C)** Growth curves of all FAST strains: the solid curve represents the mean trajectory of three independent biological replicates, and the shaded region indicates the 95% confidence interval. **(D)** Mean doubling times for each strain, calculated from the growth curves in (C); errors represent one standard deviation (SD) across three biological replicates. **(E)** mtDNA copy number (CN) of FAST strains

relative to the mtDNA$^{ATP6\text{-}NG}$ strain, determined via qPCR. qPCR experiments were performed three times, each with three biological replicates and three technical replicates per biological replicate. **(F)** Doubling time of mtDNA$^{ATP6\text{-}NG}$ cells grown on solid medium. Either 10, 20, or 30 individual cells were relocated to defined positions on the agar plate and cultured for 20 hours to allow microcolony formation. Subsequently, the resulting colonies were independently harvested into 100 μL of SC medium, and cell numbers were quantified by flow cytometry. Doubling times were calculated based on the known initial and final cell counts. Each data point represents a biological replicate. **(G)** Schematic of the workflow of the simulation used for Figs 2G and 4F. **(H)** Post-growth mixing assay of haploid Δ*atp20* cells. **(I)** mtDNA CN in Δ*atp20* cells relative to the mtDNA$^{ATP6\text{-}NG}$ strain. Panels (B),(E),(H) and (I) were analysed for statistical significance using an unpaired student T-test performed on the means of the replicates (*P <0.05; **P <0.005; ***P < 0.001).
(TIFF)

**S3 Fig. Characterization of the ratiometric ATP sensor QUEEN-2m. (A)** Maximum-intensity projections of widefield fluorescence microscopy images of a cell expressing mtQUEEN-2m and mtKate2, with corresponding **(B)** line profiles of the respective fluorescent channels. The yellow line with a white arrow in the "Merge" panel indicates the pixel positions used for intensity profiling. **(C)** Comparison of mitochondrial network length in cells expressing mtNG or mtQUEEN-2m. **(D)** Drop-dilution assay of mtDNA$^{il}$ and mtDNA$^{ic}$ cells expressing no fluorophore, mtNG, or mtQUEEN-2m. Cells were spotted onto YPD (glucose) or YPG (glycerol) plates and incubated for 48 h at either 30°C or 37°C. **(E)** Quantification of relative cytosolic ATP levels in FAST strains expressing cytQUEEN-2m, shown together with a representative ratiometric image. Scale bars in (A) and (E) represent 5 μm. Panels (C) and (E) were analysed for statistical significance using an unpaired Student's t-test performed on replicate means (*P < 0.05; **P < 0.005; ***P < 0.001).
(TIFF)

**S4 Fig. Phyisological parameters in cells lacking a functional respiratory chain. (A)** Maximum-intensity projection of a widefield fluorescence microscopy image of a Zygote. One parental cell expresses LacI-3xmNeongreen that can bind to the LacO repeats integrated in its mtDNA and mitochondrial targeted mKate2. Prior to mating its cell wall was stained with conA-Alexa633. The other cell does not express any fluorophores and its mtDNA lacks LacO repeats. Zygotes were stained with 2μg/ml DAPI to visualize the entire mtDNA pool of the cell. Note that LacI-3xmNeongreen spots are only observed in one parental cell. **(B)** Line profiles of DAPI and Laci-NG intensities of 50 Zygotes with a line width of 41 pixels as depicted in the BF panel of panel (A). Intensities were each normalized by z-score scaling **(C)** Representative widefield fluorescence micrographs of zygotes expressing mtNG (corresponding to Fig 3G). Zygotes were stained with 50nM TMRM to assess ΔΨ. Parental cells contain either the same mitochondrial genome (upper row) or comprise WT and mutant mtDNA; P1 and P2 cells are indicated in the bright-field panel. Prior to mating, P1 cell walls were labeled with conA-Alexa633. Superplots of **(D)** membrane potential levels or **(E)** mitochondrial ATP levels in zygotes of WT, Δ*rip1*, and Δ*cox4* nuclear backgrounds. The three big dots represent the mean of each replicate and small dots represent individual-cell values. Scale bar in (A) and (C) = 5μM. Panels (D) and (E) were analysed for statistical significance using an unpaired student T-test was performed on the means of the replicates (*P <0.05; **P <0.005; ***P < 0.001).
(TIFF)

**S5 Fig. Growth phenotypes and mtDNA CN of mtDNA mutant strains with additional deletions of nuclear-encoded *RIP1* or *COX4* (A)** Fraction of cells within a mtDNA$^{ATP6\text{-}NG}$ population without NG signal **(B)** Post-growth mixing assay of Δ*rip1* and Δ*cox4* nuclear-background strains compared to WT. **(C)** Drop dilution assay of mtDNA$^{ATP6\text{-}NG}$ and mtDNA$^{Δatp6}$ strains in the WT, Δ*rip1* and Δ*cox4* nuclear background. Serial dilutions were spotted on YPD and incubated for 24 h at 30 °C. **(D)** mitochondrial copy numbers relative to WT mtDNA$^{ATP6\text{-}NG}$ for Δ*rip1* and Δ*cox4* strains. Panels (A), (B) and (C) were analysed for statistical significance using an unpaired student T-test was performed on the means of the replicates (*P <0.05; **P <0.005; ***P < 0.001).
(TIFF)

**S1 Dataset. Rawdata used for graphs.** This Excel file includes all data used to generate the figures in this manuscript. (XLSX)

## Acknowledgments

We are grateful to AG Kunz and AG Robatzek for providing access to the flow cytometers. We thank Nadja Lebedeva for her technical assistance and to Verena Peterreins for cloning the mtQUEEN-2m plasmid. We also appreciate the valuable discussions and feedback from members of the Mokranjac lab and the Mito-Club.

## Author contributions

**Conceptualization:** Felix Thoma, Kurt M. Schmoller, Christof Osman.

**Data curation:** Felix Thoma, Johannes Hagen, Romina Rathberger, Francesco Padovani, David Hörl.

**Formal analysis:** Felix Thoma, Johannes Hagen, Romina Rathberger, Francesco Padovani, David Hörl.

**Funding acquisition:** Christof Osman.

**Investigation:** Felix Thoma, Johannes Hagen, Romina Rathberger.

**Methodology:** Felix Thoma, Johannes Hagen.

**Software:** Francesco Padovani, David Hörl.

**Supervision:** Kurt M. Schmoller, Christof Osman.

**Validation:** Johannes Hagen.

**Visualization:** Felix Thoma, Johannes Hagen, Francesco Padovani.

**Writing – original draft:** Felix Thoma, Francesco Padovani, Kurt M. Schmoller, Christof Osman.

**Writing – review & editing:** Felix Thoma, Johannes Hagen, Francesco Padovani, Kurt M. Schmoller, Christof Osman.

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
