## [Decision Letter · Decision Letter 0]

22 Sep 2025

PGENETICS-D-25-00904

Local Mitochondrial Physiology Defined by mtDNA Quality Guides Purifying Selection

PLOS Genetics

Dear Dr. Osman,

Thank you for submitting your manuscript to PLOS Genetics. After careful consideration, we feel that it has merit but does not fully meet PLOS Genetics's publication criteria as it currently stands. Therefore, we invite you to submit a revised version of the manuscript that addresses the points raised during the review process.

I have received three independent reports on your submission, which are attached. As you may see, all three referees found that the methodology that you have set up and, overall speaking, the conclusions of your paper, are sound and are worth of interest for the audience of PLOS Genetics and, in general, for readers interested in mitochondrial DNA inheritance and redistribution in the presence of deleterious mutations.

These positive reactions are qualified by different degrees of enthusiasm. As you may see, one referee suggests to include just some textual modifications to add discussion on the mechanistic basis of selective inheritance. A second referee, while still considering that the paper is a useful contribution to the field, suggests the addition of several controls that would be required to interpret your experimental data. This referee asks you to consider that the half-life of the Neon Green reporter, if excessively long, could distort your data. Another, in my opinion, sound point raised by this referee is that you would need to consider the possibility of second-side mutations resulting in the complete loss of mitochondrial DNA being selected among the mutant population. This referee also indicates the need of taking into account the copy number of mitochondrial DNA in the interpretation of your results and the possible changes in copy number caused by the deletion mutations. Lastly, this referee also suggests an alternative control to the Post-Growth Mixing Assay.

The third referee was concerned by the assays used to monitor the physiological state of the mitochondria. For a microscopist like myself, familiar with the aspect of mitochondria, it is pretty clear that the QUEEN probe is targeted to the mitochondria and does not measure cellular ATP levels, but specifically the mitochondrial ones. I searched the materials and methods section for a description of the mt-Queen methodology, and it seems to be missing. Actually, the only place in the manuscript in which it is clearly stated that the mt-QUEEN reporter is targeted to mitochondria is on figure 3C legend. which is potentially confusing. I am inclined to think that this aspect of the manuscript should be discussed in quite more detail and additional controls discussed (does mtQueen affect my physiology?). I should also like to ask you to please indicate the carbon source used during your dissection procedures explicitly for each individual case. Lastly, although I appreciate that this is a minor point, please take care of your wording when referring to ‘deletions of the proteins’.

Overall, please address thoroughly the comments of the referees, include a thorough description of ATP measuring techniques and if you consider it appropriate, return the manuscript to me.

With warm regards.

Miguel

Please submit your revised manuscript within 60 days Nov 21 2025 11:59PM. If you will need more time than this to complete your revisions, please reply to this message or contact the journal office at plosgenetics@plos.org. Please include the following items when submitting your revised manuscript:

We look forward to receiving your revised manuscript.

Kind regards,

Miguel A Peñalva

Academic Editor

PLOS Genetics

Pablo Wappner

Section Editor

PLOS Genetics

Aimée Dudley

Editor-in-Chief

PLOS Genetics

Anne Goriely

Editor-in-Chief

PLOS Genetics

**Journal Requirements:**

https://journals.plos.org/plosgenetics/s/submission-guidelines#loc-parts-of-a-submission

3) We have noticed that you have uploaded Supporting Information files, but you have not included a complete list of legends. Please add a full list of legends for your Supporting Information files (Supplementary Tables) after the references list. Please ensure that the Supplementary Tables are labeled as "S1 Table" and "S2 Table."

4) We notice that your supplementary figure 4 is uploaded with the file type 'Figure'. Please amend the file type to 'Supporting Information'. Please ensure that each Supporting Information file has a legend listed in the manuscript after the references list.

5) We notice that your supplementary Figures are included in the manuscript file. Please remove them from the main file of the manuscript as they should be uploaded separately with the file type 'Supporting Information'. Please ensure that each Supporting Information file has a legend listed in the manuscript after the references list.

6) In the online submission form, you indicated that "The raw microscopy data can be made available upon request." All PLOS journals now require all data underlying the findings described in their manuscript to be freely available to other researchers, either

1. In a public repository

2. Within the manuscript itself

3. Uploaded as supplementary information.

Note: Authors must share the “minimal data set” for their submission. PLOS defines the minimal data set to consist of the data required to replicate all study findings reported in the article, as well as related metadata and methods (https://journals.plos.org/plosone/s/data-availability#loc-minimal-data-set-definition).

3) The points extracted from images for analysis.

**Reviewers' comments:**

Reviewer's Responses to Questions

Reviewer #1: Mitochondrial DNA (mtDNA) is necessary for oxidative phosphorylation, and the intactness of mtDNA a critical determinant of cellular functionality. How functional mtDNA is maintained and how nonfunctional versions are selected against is a central question in mitochondrial biology. In this work by Thoma et al, new insights into this process are revealed through sets of high-quality experiments. The authors used baker´s yeast as a model system, which they have previously used to demonstrate that portions of the mitochondrial reticulum adjacent to specific mtDNA are equipped specifically with gene products derived from these mtDNA molecules, reflecting spheres of influence that can determine selection of mtDNA molecules (PMID: 34516914). In the current work, they now established a new less-work requiring method, FAST, which allows to determine the dynamics of heteroplasmy, ie the relative amount of wild type and mutant mtDNA in the same cell. Using this and previously established assays, they very clearly demonstrate that mutant mtDNA is actively selected against, even under conditions of fermentation, when cells do not rely on respiration for energy conversion. However, when cells lack the propensity for respiration entirely, selection is no longer possible, demonstrating that oxidative phosphorylation is the principal aspect of selection. In subsequent experiments, they reveal that different portions of the mitochondrial network have different bioenergetic characteristics, presumably generating the basis for selection according to the sphere-of-influence model.

Overall, this is a very well controlled study with high-quality data sets that shed new light on an important biological process. The study is well written. While I think that additional experiments are not necessary at this stage, it would be great if the authors could in greater depth discuss possible molecular mechanisms of selection. They discuss that mitochondrial transport into the bud could be one possible way to exclude mutant mtDNA from inheritance. However, as suggested in a recent study (PMID: 40911417), it is also possible that the improved bioenergetic state around intact mtDNA could support replication of this DNA, while bioenergetic problems surrounding mutant mtDNA could lead to lower efficiency of replication and hence a competitive disadvantage.

Reviewer #2: In their manuscript, the authors use a set of S. cerevisiae mtDNA mutants to generate and study heteroplasmic cells (zygotes and their descendants). This work continues their previous studies (references 10 and 12 in the manuscript); however, in this study, they leverage the power of flow cytometry to assess the levels of mitochondria-encoded fluorescent protein in heteroplasmic cells. Furthermore, they investigate whether deletion of the ATP synthase gene ATP6 has different effects on mtDNA quality control compared to deletion of respiratory chain genes they have discovered previously. They found that the effects of ATP6 deletion are more pronounced at the cellular level (Fig 2E) than those of COB and COX2 gene deletions. Meanwhile, mtDNAs with deletions in any of these three genes show decreased intracellular fitness in heteroplasmic cells. Overall, I am rather happy with the manuscript as it stands. It was interesting to read, but I have some questions\concerns and suggest considering several additional controls.

— What is the half-life of ATP6-NG? This question is crucial for the entire study because if ATP6-NG has a sufficiently long half-life, the intensity of the NG signal can be explained by inheritance of the protein from mother cells, retained even several generations after the mtDNA ATP6-NG was displaced by mtDNA without the fluorescent protein. It should be noted that mitochondrial inner membrane proteins usually have extremely long half-lives. I suggest assessing the ATP6-NG half-life by incubating cells in the presence of a mitochondrial translation inhibitor (e.g., erythromycin) and analyzing the distribution of signal levels across the population at different time points.

— As an addition to the previous question, I would like to point to Figure 1B, which shows the absence of even a small subpopulation of NG-negative cells, while W303 strain used in this study usually contains 2–5% petite cells in suspension, which are likely unable to perform mitochondrial protein synthesis. Why is there no such population?

— How stable is mtDNA with deletions of ATP6, COB, and COX2? Secondary deletions can emerge in yeast rho− cells; could this be the case for these strains? It is also theoretically possible that rho− cells lose their mtDNA completely and become rho0. This process is favored because dysfunctional mtDNA no longer provides a fitness advantage to the host cell, and the complete loss of mtDNA can even free up resources required for mtDNA replication. I suggest checking the percentage of rho0 cells (e.g., those with DAPI-negative cytoplasm) in strains harboring deletions in mtDNA. Otherwise, the increase in NG-positive cells observed in mtDNA-ATP6NG X mtDNA deltaCOB crossings could be explained by a large fraction of rho0 cells in mtDNA deltaCOB (and in other deletion strains in a similar way). Although, Figure 4E shows that deletion of nuclear-encoded OXPHOS genes removes the difference between mtDNA-ATP6NG X mtDNA deltaCOB and mtDNA-ATP6NG X mtDNA WT crossings, it would be better to verify this with a direct control.

— Does deletion of COX2, ATP6, and COB affect the copy number of mtDNA? If these deletions reduce mtDNA levels in haploid cells, this could explain the main result without invoking the quality control mechanism hypothesis. The mtDNA copy number should be assessed because a pronounced difference between the studied strains would challenge the main conclusion of the study.

— To verify that the observed effects are not due to differences in cell-level fitness of the mutant variants, the authors performed a “post-growth mixing assay” (Figure 2C, E). However, the conditions in the mixed and clonal cultures of wild-type and OXPHOS-deficient cells might differ significantly. I wonder why the authors did not mix mtDNAATP6-NG cells and “Dark” cells prior to their growth on the plate (using same mating type cells). This approach would more closely mimic the conditions of the heteroplasmic zygote experiment.

— Figure S4-A the fact that delta rip1 and delta cox4 shows statistical significant difference in NG signal is confusing. Can it be a mistake?

Reviewer #3: The manuscript addresses the mechanisms underlying the maintenance of intact mtDNA in Saccharomyces cerevisiae and introduces an approach to analyze the segregation of mtDNA variants with potential impact on mitochondrial physiology. Given this reviewer’s scientific background, particular attention is directed toward the bioenergetic analysis of mutants affecting the OXPHOS system and the appropriateness of the protocols employed.

The analysis of mitochondrial function through measurements of mitochondrial membrane potential (MMP) using the probe TMRM, with appropriate correction for mitochondrial content, appears sound. The probe’s response is validated with CCCP controls. The existence of regions within the mitochondrial network that display varying MMP is well established in the literature. In this context, the authors demonstrate that such differences in membrane MMP arise from the spatial heterogeneity caused by the unequal distribution of mtDNA variants in heteroplasmic cells.

However, this reviewer considers unsound the evaluation of OXPHOS mutants based on measurements of cellular ATP levels using the probe QUEEN-2. In most cells, phosphorylation potential (ΔGp) is sustained by both cytosolic and mitochondrial ATP production. In contrast, S. cerevisiae grown in high glucose undergoes the Crabtree effect, where the high glycolytic flux (ethanol fermentation) should maintain ATP levels independently of OXPHOS. Indeed, the authors clearly show that all mtDNA mutants exhibit identical growth rates, which strongly indicates the absence of ATP limitations that could reduce ΔGp. Measurements of cellular ATP levels under fermentative conditions cannot be used to draw conclusions about OXPHOS function.

This reviewer does not have the expertise to assess the genetic aspects of the work in detail. Nevertheless, the study appears solid, even if conclusions regarding mtDNA heterogeneity rely exclusively on spatial variations in MMP within the network. This constitutes a critical point of consideration.

Accordingly, this reviewer recommends that the authors remove the data related to ATP measurements and resubmit the manuscript for publication. The study otherwise presents valid and substantial work.

**Have all data underlying the figures and results presented in the manuscript been provided?**

Reviewer #1: Yes

Reviewer #2: None

Reviewer #3: Yes

PLOS authors have the option to publish the peer review history of their article (what does this mean?). If published, this will include your full peer review and any attached files.

Reviewer #1: No

Reviewer #2: No

Reviewer #3: No

**Figure resubmission:**
---

## [Decision Letter · Decision Letter 1]

29 Dec 2025

Dear Dr Osman,

We are pleased to inform you that your manuscript entitled "Local Mitochondrial Physiology Defined by mtDNA Quality Guides Purifying Selection" has been editorially accepted for publication in PLOS Genetics. Congratulations!

Yours sincerely,

Miguel A Peñalva

Academic Editor

PLOS Genetics

Pablo Wappner

Section Editor

PLOS Genetics

Aimée Dudley

Editor-in-Chief

PLOS Genetics

Anne Goriely

Editor-in-Chief

PLOS Genetics

BlueSky: @plos.bsky.social

Comments from the reviewers (if applicable):

Reviewer's Responses to Questions

**Comments to the Authors:**

Reviewer #1: The authors have improved the manuscript according to the suggestions.

Reviewer #2: In the revised version of the manuscript, the authors performed additional control experiments and addressed my questions. I think the manuscript still lacks a paragraph in the Discussion that addresses the study's limitations from a methodological perspective.. However, I believe this should be at the authors' (or editor’s) discretion at this stage, and I recommend acceptance.

**Have all data underlying the figures and results presented in the manuscript been provided?**

Reviewer #1: Yes

Reviewer #2: None

PLOS authors have the option to publish the peer review history of their article (what does this mean?). If published, this will include your full peer review and any attached files.

Reviewer #1: No

Reviewer #2: No

**Data Deposition**

http://datadryad.org/submit?journalID=pgenetics&manu=PGENETICS-D-25-00904R1

**Press Queries**

---

## [Editor Report · Acceptance letter]

PGENETICS-D-25-00904R1

Local Mitochondrial Physiology Defined by mtDNA Quality Guides Purifying Selection

Dear Dr Osman,

We are pleased to inform you that your manuscript entitled "

Local Mitochondrial Physiology Defined by mtDNA Quality Guides Purifying Selection" has been formally accepted for publication in PLOS Genetics! Your manuscript is now with our production department and you will be notified of the publication date in due course.

With kind regards,

Anita Estes

PLOS Genetics

On behalf of:
